# SCHEDULENET: LEARN TO SOLVE MULTI-AGENT SCHEDULING PROBLEMS WITH REINFORCEMENT LEARNING

## ABSTRACT

We propose ScheduleNet, an RL-based decentralized constructive scheduler for coordinating multi-agent to finish tasks with minimum completion time. We formulate multi-agent scheduling problems (mSPs) as an event-based Markov decision process (MDP) with an episodic reward (e.g., makespan) and derive a decentralized decision-making policy using reinforcement learning. The decision making procedure of ScheduleNet includes: (1) representing the state of a scheduling problem with the agent-task graph, (2) extracting node embeddings for agents and tasks by employing the type-aware graph attention (TGA), and (3) computing the assignment probability with the computed node embeddings. We validate the effectiveness of ScheduleNet on two types of mSPs: multiple traveling salesmen problem (mTSP) and job-shop scheduling problem (JSP). We empirically show that ScheduleNet can outperform other heuristic approaches and existing deep RL approaches, particularly validating its exceptional effectiveness in solving large and practical problems. Furthermore, we have demonstrated that ScheduleNet can effectively solve online vehicle routing problems where the new target customer appears dynamically during the course of scheduling.

## 1 INTRODUCTION

Optimal assignments of multiple autonomous agents for sequential completion of distributed tasks are necessary to solve various types of scheduling problems in the logistics, transportation, and manufacturing industries. Examples of such scheduling problem include finding the optimal delivery plans for vaccines, customer pickup orders for ride-sharing services, and machine operation sequence in modern manufacturing facilities. As the size of the problems increases, solving large-scale scheduling problems using mathematical programming becomes infeasible or ineffective due to the expensive computational cost. Furthermore, such optimization approaches cannot solve real-time scheduling problems where the new tasks appear dynamically.

**Target Problem & Challenges**. In this paper, we propose ScheduleNet, an RL-based decentralized constructive scheduler for coordinating multiple agents to finish tasks with minimum completion time. The objectives of the target problems and their associated technical challenges are as follows:

- *Min-Max vs Min-Sum*: ScheduleNet seeks to minimize the total completion time (i.e., makespan) for various time-critical distributed tasks (e.g., vaccine delivery). Most multi-agent scheduling problems are designed to minimize the total traveling distance of all agents, which often result in unbalanced task assignments among the agents (Bakach et al., 2021). Although the makespan is the most direct and intuitive reward for inducing coordination, it is notoriously difficult to train the decentralized policy with this sparse and delayed reward due to the temporal and spatial credit assignment issues.

- *Construction Heuristic vs Improvement heuristic*: ScheduleNet builds a solution sequentially by assigning an idle agent to one of the remaining tasks while considering the relationships among the remaining tasks and agents. This construction heuristic ensures that the learned policy can reschedule whenever a new event occurs (i.e., an agent finishes the assigned task or a new task appears). However, it is more challenging to find a better plan due to its sequential solution construction; a wrong choice at the early stage can cause irreversible poor results at the end.

- **Decentralized** *vs Centralized*: ScheduleNet allows each agent to choose its destination independently while using its local observations and incorporating other agents' assignments. This decentralization ensures that the learned policy can solve a large-scale problem without having to search over the joint action space for all agents. However, to make the independently-chosen scheduling decision produce an excellent global performance, a sophisticated coordination mechanism should be incorporated into decentralized policy implicitly.

**Proposed Decision-Making Scheme**. We formulate the multi-agent scheduling problems (mSPs) as an event-based Markov decision process (MDP) with an episodic reward, and derive a decentralized decision-making policy using reinforcement learning. At every step, ScheduleNet accepts the MDP state as an input and assigns an idle agent to one of the feasible tasks. The decision-making procedure of ScheduleNet is as follows:

- ScheduleNet first represents the MDP state as an agent-task graph, which captures the complex relationships among the entities effectively and is general enough to be applied to various mSPs.
- ScheduleNet then employs the type-aware graph attention (TGA) to extract important relational features among the agents and tasks for making the best cooperative task assignment.
- Lastly, ScheduleNet computes the agent-task assignment probability by utilizing the computed node embeddings.

**Training Method**. Although the makespan (shared team reward) is the most direct and general reward design for solving mSPs, training a decentralized scheduling policy using this reward is extremely difficult due to the credit assignment issues (Riedmiller et al., 2018; Hare, 2019). Additionally, makespan is highly volatile due to the combinatorial aspect of mSPs' solution space; a slight change in the solution can drastically alter the outcome. To overcome these issues, we employ the Clip-REINFORCE algorithm with normalized reward to train the decentralized cooperative policy effectively.

**Novelties**. The proposed method that derives the decentralized constructive schedulers to coordinate multiple agents has the following novelties and advantages:

- *Balance between Representability* & *Scalability*: ScheduleNet can extract crucial features effectively using TGA and make the best cooperative task assignment (representability) in a computationally efficient manner. The computationally-efficient representation scheme and the constructive decision-making scheme allows ScheduleNet to solve large-scale scheduling problems.
- *Transferability* & *Generalizability*: The type-aware graph representation allows the trained policy to solve problems with different numbers of agents and tasks (size transferability). Furthermore, this general state representation with the universal reward signal (i.e., makespan) allows ScheduleNet to be used to solve various multi-agent scheduling problems (generalizability). We validate this by showing that ScheduleNet can learn to solve multiple traveling salesmen problem (mTSP), as well as jop-shop scheduling problem (JSP) whose constraints are more complex than that of mTSP.

## 2 RELATED WORKS

**RL approaches that solve vehicle routing problems**. According to Mazyavkina et al. (2020), the RL approaches that solve vehicle routing problems can be categorized into: (1) the improvement heuristics that rewrite the complete solution iteratively to obtain a better routing plan (Wu et al., 2020; da Costa et al., 2020; Chen & Tian, 2019; Lu et al., 2020); (2) the construction heuristics that construct the solution sequentially by assigning idle vehicles to unvisited cities until the complete routing plan (sequence) is constructed (Bello et al., 2016; Nazari et al., 2018; Kool et al., 2018; Khalil et al., 2017), and (3) the hybrid approaches that blend both approaches (Joshi et al., 2020; Fu et al., 2021; Kool et al., 2021; Ahn et al., 2020). Typically, the improvement heuristics show better performances than construction heuristics as they revise the complete plan iteratively. However, the construction heuristics are more effective for online vehicle routing problems, where the routes should be updated whenever a new customer appears. These RL approaches have exclusively focused on static planning in a single-agent perspective, which is far from the settings of real applications.

**RL approaches that solve min-max mTSP**. There are only few RL approaches that solve min-max mTSP, which involves minimizing the makespan for multiple salesmen to visit all cities. Hu et al. (2020) applies RL to train the clustering algorithm that groups cities, and strong TSP heuristics (e.g., OR-Tool) to optimize the sub-tours of the city clusters. This is fundamentally different from ScheduleNet, which derives a complete end-to-end learned heuristic that constructs a feasible solution from "scratch" without relying on any existing solvers. Cao et al. (2021), which proposes a transformer-based construction policy to solve min-max mTSP, is the most similar approach to ScheduleNet.

**RL approaches that solve Job-shop scheduling problems**. There are only few RL approaches that solve the JSPs, which involves minimizing the makespan for multiple machines to finish the sequence of operations that is required to finish their jobs. For example, Gabel & Riedmiller (2012); Lin et al. (2019) propose to learn a scheduling policy for each machine; hence, it requires additional training to solve JSPs with a different number of machines. Recently, Park et al. (2021); Zhang et al. (2020) have proposed to learn a shared scheduling policy that can be used for all machines to minimize the makespan. Similar to ScheduleNet, these studies utilize the disjunctive graph representation of JSP. However, these methods utilize human-engineered dense reward, while ScheduleNet uses sparse and delayed reward (i.e., makespan) to train a policy.

## 3 PROBLEM FORMULATION

We formulate mSP as a MDP with sparse reward, and aim to derive a decentralized scheduling decision-making policy that can be shared by all agents. The MDP is defined as:

**State**. We define state $s_\tau$ as the $\tau$-th partial solution of mSP (i.e., the completed/uncompleted tasks, the status of agents, and the sequence of the past assignments). The initial $s_0$ and terminal state $s_T$ are defined as an empty and a complete solution, respectively.

**Action**. We define action $a_\tau$ as the act of assigning an idle agent to one of the feasible tasks (unassigned tasks). We refer to $a_\tau$ as the *agent-to-task assignment*. When the multiple agents are idle at the same time $t$, we randomly choose one agent and assign an action to the agent. This is repeated until no agent is idle. Note that such randomness does not alter the resulting solutions, since the agents are considered to be homogeneous and the scheduling policy is shared.

**Transition**. The proposed MDP is formulated with an *event*-based transition. An event is defined as the case where any agent finishes the assigned task (e.g., a salesman reaches the assigned city in mTSP). Whenever an event occurs, the idle agent is assigned to a new task, and the status of the agent and the target task are updated accordingly. We enumerate the event with $\tau$ to avoid confusion from the elapsed time of the problem; $t(\tau)$ is a function that returns the time of event $\tau$.

**Reward**. The proposed MDP uses the negative makespan (i.e. total completion time of tasks) as the reward (i.e., $r(s_T) = -t(T)$) that is realized only at $s_T$.

### 3.1 EXAMPLE: MDP FORMULATION OF MTSP

Let us consider the single-depot mTSP with two types of entities: $m$ salesmen (i.e., $m$ agents) and $N$ cities (i.e., $N$ tasks). All salesmen start their journey from the depot, and come back to the depot after visiting all cities (each city can be visited by only one salesman). The solution to mTSP is considered to be *complete* when all the cities have been visited, and all salesmen have returned to the depot. The MDP formulation for mTSP is similar to that of the general mSP. The specific definition of the state for mTSP is as follows:

**State.** We define $s_\tau = (\{s_\tau^i\}_{i=1}^{N+m}, s_\tau^{\text{env}})$, which is composed of two types of states: entity state $s_\tau^i$ and environment state $s_\tau^{\text{env}}$.

- $s_\tau^i = (p_\tau^i, \mathbf{1}_\tau^{\text{active}}, \mathbf{1}_\tau^{\text{assigned}})$ is the state of the $i$-th entity. $p_\tau^i$ is the 2D Cartesian coordinate of the $i$-th entity at the $\tau$-th event. $\mathbf{1}_\tau^{\text{active}}$ indicates whether the $i$-th agent/task is active (agent is working/ task is not visited). Similarly, $\mathbf{1}_\tau^{\text{assigned}}$ indicates whether agent/task is assigned.
- $s_\tau^{\text{env}}$ contains the current time of the environment, and the sequences of cities visited by the salesmen.

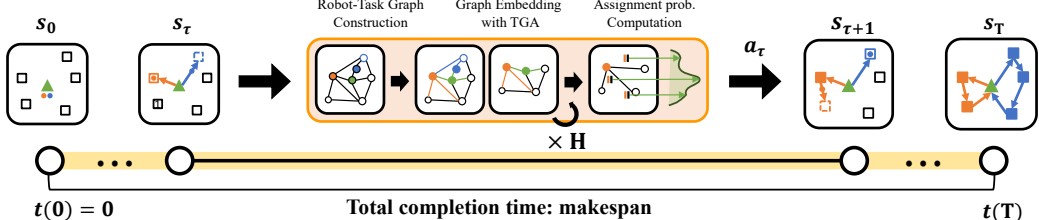

**Figure 1: Solving mSP with ScheduleNet.** At every event of the MDP, ScheduleNet constructs the agent-task graph $\mathcal{G}_\tau$ from $s_\tau$, then computes the node embedding of $\mathcal{G}_\tau$ using TGA, and finally computes the agent-task assignment probabilities from the node embedding.

## 4 SCHEDULENET

In this section, we explain how ScheduleNet recommends a scheduling action $a_\tau$ of an idle agent from input $s_\tau$ (partial solution). This is done by (1) constructing the agent-task graph $\mathcal{G}_\tau$, (2) embedding $\mathcal{G}_\tau$ using TGA, and (3) computing the assignment probabilities. Figure 1 illustrates the decision-making process of ScheduleNet.

### 4.1 CONSTRUCTING AGENT-TASK GRAPH

ScheduleNet constructs the *agent-task graph* $\mathcal{G}_\tau$ that reflects the complex relationships among the entities in $s_\tau$. Specifically, ScheduleNet constructs a directed complete graph $\mathcal{G}_\tau = (\mathbb{V}, \mathbb{E})$ out of $s_\tau$, where $\mathbb{V}$ is the set of nodes and $\mathbb{E}$ is the set of edges. The nodes and edges, and their associated features are defined as:

- $v_i$ denotes the $i$-th node, which represents either an agent or a task. $v_i$ contains the node feature $x_i = (s_\tau^i, k_i)$, where $s_\tau^i$ is the state of entity $i$, and $k_i$ is the type of $v_i$. For example, if the entity $i$ is an agent and $\mathbf{1}_\tau^{active} = 1$, then $k_i$ becomes *active-agent*. For the full list of the types, refer to Appendix C.2.

- $e_{ij}$ denotes the edge between the source node $v_j$ and the destination node $v_i$. The edge feature $w_{ij}$ is equal to the Euclidean distance between the two nodes.

In the following subsections, we omit the event iterator $\tau$ for notational brevity, since the action selection procedure is only associated with the current event index $\tau$.

### 4.2 GRAPH EMBEDDING USING TGA

ScheduleNet computes the node embeddings from the agent-task graph $\mathcal{G}$ using the TGA, which is designed to capture the different relations among the graph entities by applying attention mechanisms for each relational type. TGA uses three steps to compute the updated node/edge embedding as follows:

**Type-aware edge update**. Given the node embedding $h_i$ and edge embedding $h_{ij}$, TGA computes the type-aware edge embedding $h'_{ij}$ and the attention logit $z_{ij}$ as follows:

$$
\begin{aligned}
h'_{ij} &= \mathrm{TGA}_\mathbb{E}([h_i, h_j, h_{ij}], k_j) \\
z_{ij} &= \mathrm{TGA}_\mathbb{A}([h_i, h_j, h_{ij}], k_j)
\end{aligned}
\tag{1}
$$

where $\mathrm{TGA}_\mathbb{E}$ and $\mathrm{TGA}_\mathbb{A}$ are the type-aware edge update function and the type-aware attention function, respectively. $\mathrm{TGA}_\mathbb{E}$ and $\mathrm{TGA}_\mathbb{A}$ are parameterized using multilayer perceptrons (MLPs).

**Type-aware message aggregation**. Each entity in the agent-task graph interacts differently with the other entities, depending on the type of the edge between them. To preserve the different relationships among the entities during the graph embedding procedure, TGA gathers messages $h'_{ij}$ via the type-aware message aggregation.

First, TGA aggregates messages for each node type and produces the *per-type* message $m_i^k$ as follows:

$$m_i^k = \sum_{j \in \mathcal{N}_k(i)} \alpha_{ij} h'_{ij} \tag{2}$$

where $\mathcal{N}_k(i) = \{v_l | k_l = k, v_l \in \mathcal{N}(i)\}$ is the type $k$ neighborhood of $v_i$, and $\alpha_{ij}$ is the attention score that is computed using $z_{ij}$:

$$\alpha_{ij} = \frac{\exp(z_{ij})}{\sum_{j \in \mathcal{N}_k(i)} \exp(z_{ij})} \tag{3}$$

TGA aggregates the per-type messages to compute the total aggregated message $m_i$ for $v_i$ as:

$$m_i = \sum_{k \in \mathbb{K}} m_i^k \tag{4}$$

where $\mathbb{K}$ is the set of node types.

**Type-aware node update**. The total aggregated message $m_i$ is then used to compute the updated node embedding $h'_i$ as follows:

$$h'_i = \text{TGA}_\mathbb{V}([h_i, m_i], k_i) \tag{5}$$

where $\text{TGA}_\mathbb{V}$ is the type-aware node update function that is parametrized using MLP. The detailed architectures of $\text{TGA}_\mathbb{E}$, $\text{TGA}_\mathbb{A}$, and $\text{TGA}_\mathbb{V}$ are provided in Appendix A.

ScheduleNet computes the node embeddings from $\mathcal{G}$ using TGA. The embedding procedure first encodes the features of $\mathcal{G}$ into the initial node embeddings $\{h_i^{(0)} | v_i \in \mathbb{V}\}$, and the initial edge embeddings $\{h_{ij}^{(0)} | e_{ij} \in \mathbb{E}\}$. ScheduleNet then performs TGA $H$ times on all nodes to compute the final node embeddings $\{h_i^{(H)} | v_i \in \mathbb{V}\}$ and edge embeddings $\{h_{ij}^{(H)} | e_{ij} \in \mathbb{E}\}$.

### 4.3 COMPUTING ASSIGNMENT PROBABILITY

Using $\{h_i^{(H)} | v_i \in \mathbb{V}\}$ and $\{h_{ij}^{(H)} | e_{ij} \in \mathbb{E}\}$, ScheduleNet selects the assignment action $a_\tau$ for the target idle agent. It computes the assignment probability of the target agent $i$ to the *unassigned* task $j$ as follows:

$$\begin{aligned} l_{ij} &= \text{MLP}_{actor}(h_i^{(H)}, h_j^{(H)}, h_{ij}^{(H)}) \\ p_{ij} &= \text{softmax}(\{l_{ij}\}_{j \in \mathbb{A}(\mathcal{G}_\tau)}) \end{aligned} \tag{6}$$

where $\mathbb{A}(\mathcal{G}_\tau)$ denotes a set of feasible actions that is defined as $\{v_j | k_j = \text{Unassigned-task}, v_j \in \mathbb{V}\}$.

Note that ScheduleNet allows an agent to process its local state information and make the assignment choice in a decentralized way. This enables ScheduleNet to solve mSPs with any number of agents and tasks even when the agents or tasks appear dynamically.

## 5 TRAINING SCHEDULENET

We utilize the sparse team reward (makespan) as the reward to train the decentralized scheduler (ScheduleNet) that aims complete the tasks as quickly as possible by coordinating multiple agents. Even though this team reward is the most direct signal that can be used for solving various types of mSPs, training a decentralized cooperative policy using a single sparse and delayed reward is notoriously difficult (Riedmiller et al., 2018; Hare, 2019). The high variance of the reward, due to the combinatorial nature of mSP, adds an additional difficulty. To handle such difficulties, we employ two training stabilizers, reward normalization, and Clip-REINFORCE.

### 5.1 REWARD NORMALIZATION

We denote the makespan induced by policy $\pi_\theta$ as $M(\pi_\theta)$. We observe that $M(\pi_\theta)$ is highly volatile depending on the problem size $(N, m)$ and $\pi_\theta$. To reduce the variance of the reward incurred from

the problem size, we propose to use the normalized makespan $\bar{M}(\pi_\theta, \pi_b)$ computed as:

$$\bar{M}(\pi_\theta, \pi_b) = \frac{M(\pi_\theta) - M(\pi_b)}{M(\pi_b)} \tag{7}$$

where $\pi_\theta, \pi_b$ is the current policy and baseline policy respectively.

A similar normalization scheme that only measures the performance difference between $\pi$ and $\pi_b$ has been investigated in RL applications for solving single-agent scheduling problems (Kool et al., 2018; Cao et al., 2021; Kwon et al., 2020). Such normalization can provide consistent learning signals when the training instance sizes (i.e. $N$) are fixed, which is a common practice for training RL methods to solve single-agent scheduling problems. However, we observed that even if $N$ is fixed, the (optimal) makespan of mTSP can differ severely as $m$ changes. To reduce the variability of the makespan due to $m$, we further divide the makespan difference by $M(\pi_b)$.

Using $\bar{M}(\pi_\theta, \pi_b)$, we compute the normalized return $G_\tau(\pi_\theta, \pi_b)$ as follows:

$$G_\tau(\pi_\theta, \pi_b) \triangleq -\gamma^{T-\tau} \bar{M}(\pi_\theta, \pi_b) \tag{8}$$

where $T$ is the index of the terminal state, and $\gamma$ is the discount factor of MDP. The minus sign is for minimizing the makespan. Note that, in the early phase of mSP (when $\tau$ is small), it is difficult to estimate the makespan. Thus, we place a smaller weight (i.e, $\gamma^{T-\tau}$) on $\bar{M}(\pi_\theta, \pi_b)$, which is evaluated when $\tau$ is small (early stage).

## 5.2 CLIP-REINFORCE

Even a small change in a single assignment can result in a dramatic change to the makespan due to the combinatorial nature of mSP. Hence, training the value function that predicts $G_\tau$ reliably is difficult. We thus propose to utilize Clip-REINFORCE, a variant of PPO (Schulman et al., 2017) *without* the learned value function, for training ScheduleNet. The objective of the Clip-REINFORCE is given as follows:

$$\mathcal{L}(\theta) = \mathop{\mathbb{E}}_{(\mathcal{G}_\tau, a_\tau) \sim \pi_\theta} [\min(\text{clip}(\rho_\tau, 1 - \epsilon, 1 + \epsilon)G_\tau, \rho_\tau G_\tau)] \tag{9}$$

where $\text{clip}(x, a, b) = \{a \text{ if } x \leq a, x \text{ if } a < x < b, b \text{ if } x \geq b\}$, $\epsilon$ is the clipping parameter, $G_\tau$ is a shorthand notation for $G_\tau(\pi_\theta, \pi_b)$, and $\rho_\tau = \pi_\theta(a_\tau | \mathcal{G}_\tau) / \pi_{\theta_{old}}(a_\tau | \mathcal{G}_\tau)$ is the ratio between the current policy $\pi_\theta$ and the policy before the update $\pi_{\theta_{old}}$.

## 6 EXPERIMENTS

In this section, we evaluate the performance of ScheduleNet on mTSP and JSP. To calculate the inference time, we run all experiements on the server equipped with AMD Threadripper 2990WX CPU. We use single CPU core for evaluating all algorithms.

### 6.1 MTSP EXPERIMENTS

**Training.** We denote $(N \times m)$ as the mTSP with $N$ cities (tasks) and $m$ salesmen (agents). To generate a random mTSP instance, we sample $N$ and $m$ from $U(15, 30)$ and $U(3, 4)$, respectively. Similarly, the Euclidean coordinates of $N$ cities are sampled uniformly from the unit square. ScheduleNet is trained on random mTSP instances that are generated on-the-fly. For all mTSP experimental results, we evaluate the performance of a single trained ScheduleNet model. Please refer to Appendix C.3 for more information regarding the training details.

**Results on random mTSP datasets**. To investigate the generalization capacity of ScheduleNet to various problem sizes, we evaluate ScheduleNet on the randomly generated mTSP datasets. Each $(N \times m)$ dataset consists of 100 random uniform mTSP instances.

We consider the four baseline algorithms to measure the performance of ScheduleNet. As the non-learning baselines, LKH3 (Helsgaun, 2017) and Google OR-Tools (Perron & Furnon) are used. It is noteworthy that LKH3 is known to find the optimal solutions for the mTSP problems with the identified optimal solutions. Thus, we use the makespans computed by LKH3 as the proxies for the

**Table 1: Random mTSP results.** GNN-DisPN results are reproduced from Hu et al. (2020).

| $N$ | 30 | | 50 | | | 100 | | | 200 | | | Gap |
|---|---|---|---|---|---|---|---|---|---|---|---|---|
| $m$ | 3 | 5 | 5 | 7 | 10 | 5 | 10 | 15 | 10 | 15 | 20 | |
| LKH3 | 2.17 | 1.94 | 2.00 | 1.95 | 1.91 | 2.20 | 1.97 | 1.98 | 2.04 | 2.00 | 1.97 | 1.00 |
| OR-Tools | 2.25 | 1.95 | 2.04 | 1.96 | 1.91 | 2.41 | 2.03 | 2.03 | 2.33 | 2.33 | 2.37 | 1.07 |
| GNN-DisPN | - | - | 2.12 | - | 1.95 | 2.48 | 2.09 | - | - | - | - | - |
| DAN (g.) | 2.32 | 2.02 | 2.29 | 2.11 | 2.03 | 2.72 | 2.17 | 2.09 | 2.40 | 2.20 | 2.15 | 1.13 |
| DAN (s.64) | 2.22 | 1.96 | 2.12 | 1.99 | 1.95 | 2.55 | 2.05 | 2.00 | 2.40 | 2.20 | 2.15 | 1.08 |
| ScheduleNet (g.) | 2.32 | 2.02 | 2.17 | 2.07 | 1.98 | 2.59 | 2.13 | 2.07 | 2.45 | 2.24 | 2.17 | 1.09 |
| ScheduleNet (s.64) | 2.22 | 1.96 | 2.07 | 1.99 | 1.92 | 2.43 | 2.03 | 1.99 | 2.25 | 2.08 | 2.05 | 1.04 |

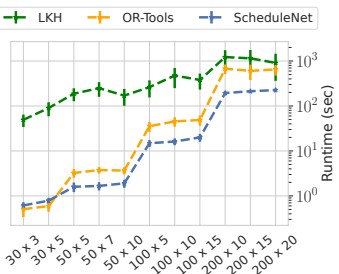

**Figure 2:** mTSP runtimes

**Table 2: mTSPLib results.** CPLEX results with ∗ are optimal solutions. Otherwise, the known-best upper bound of CPLEX results are reported.

| Instance | eil51 | | | | berlin52 | | | | eil76 | | | | rat99 | | | | Gap |
|---|---|---|---|---|---|---|---|---|---|---|---|---|---|---|---|---|---|
| $m$ | 2 | 3 | 5 | 7 | 2 | 3 | 5 | 7 | 2 | 3 | 5 | 7 | 2 | 3 | 5 | 7 | |
| CPLEX | 222.7* | 159.6 | 124.0 | 112.1 | 4110.2 | 3244.4 | 2441.4 | 2440.9 | 280.9* | 197.3 | 150.3 | 139.6 | 728.8 | 587.2 | 469.3 | 443.9 | 1.00 |
| LKH3 | 222.7 | 159.6 | 124.0 | 112.1 | 4110.2 | 3244.4 | 2441.4 | 2440.9 | 280.9 | 197.3 | 150.3 | 139.6 | 728.8 | 587.2 | 469.3 | 443.9 | 1.00 |
| OR-Tools | 243.3 | 170.5 | 127.5 | 112.1 | 4665.5 | 3311.3 | 2482.6 | 2440.9 | 318.0 | 212.4 | 143.4 | 128.3 | 762.2 | 552.1 | 473.7 | 442.5 | 1.03 |
| DAN (g) | 274.2 | 178.9 | 158.6 | 118.1 | 5,226.0 | 4,278.4 | 2,758.8 | 2,696.8 | 361.1 | 251.5 | 170.9 | 148.5 | 930.8 | 674.1 | 504.0 | 466.4 | 1.18 |
| DAN (s. 64) | 252.9 | 178.9 | 128.2 | 114.3 | 5,097.7 | 3,455.7 | 2,677.1 | 2,494.5 | 336.7 | 228.1 | 157.9 | 134.5 | 966.5 | 697.7 | 495.6 | 462.0 | 1.11 |
| ScheduleNet (g.) | 263.9 | 200.5 | 131.7 | 116.9 | 4,826.1 | 3,644.2 | 2,757.8 | 2,514.6 | 330.2 | 228.8 | 163.9 | 144.4 | 843.8 | 691.8 | 524.3 | 480.8 | 1.13 |
| ScheduleNet (s.64) | 239.3 | 173.5 | 125.8 | 112.2 | 4,591.6 | 3,276.1 | 2,517.3 | 2,441.4 | 317.7 | 220.8 | 153.8 | 131.7 | 781.2 | 627.1 | 502.3 | 464.4 | 1.05 |

optimal solutions. As the RL baselines, GNN-DisPN (Hu et al., 2020) and DAN (Cao et al., 2021) are considered (To the best of our knowledge, these are the only RL-based algorithms tried to solve the min-max mTSP problems).

Table 1 shows the average makespans and average gaps (i.e., the relative makespan w.r.t LKH3) of ScheduleNet and the baseline algorithms on various-sized random mTSP instances. ScheduleNet generally shows leading performances compared to the baseline algorithms even though it is trained on the smallest test setup (30×3). In addition, Figure 2 compares the average computation times and their standard deviations for each problem size, clearly indicating that ScheduleNet is significantly faster than traditional heuristic solvers (LKH3 and OR-Tools). The computational time difference among RL-approaches is almost not noticeable.

**Results on Public Benchmarks**. Next, to explore the generalization of ScheduleNet to problems that come from completely different distributions (e.g., real-world data), we present the results on the mTSPLib dataset defined by Necula et al. (2015). mTSPLib consists of four instances of size 51, 52, 76 and 99 from TSPLib (Reinelt, 1991), each of which is extended to multi-agent setups where $m$ equals to 2, 3, 5, and 7.

Table 2 shows the the makespans and average gaps w.r.t CPLEX of the algorithms. ScheduleNet still exhibits comparable performance to OR-tools, while DAN shows significant performance drops. From this experiment, we can observe that ScheduleNet is effective in solving both randomly generated and real-world mTSPs problems (i.e., robust transferability over data distribution shift).

## 6.2 ABLATION STUDIES

In this paper, we propose TGA as the representation learning module, and Clip-REINFORCE (CR) as the training method. Here, we provide the experimental results that show the effect of proposed TGA and CR to the training and testing performance of ScheduleNet.

We train variants of the ScheduleNet models as the target of the ablation studies as follows:

- `TGA-REINFORCE`: the model using TGA and is trained by REINFORCE (Williams, 1992).

- `GN-CR`: the model using the graph network (GN) layer (Battaglia et al., 2018) is trained by CR.

- `TGA-CR`: the model using TGA and is trained by CR; the proposed ScheduleNet.

Figure 3 compares the normalized makespans of the three ablation models on the validation test instance (30×3) during training, and Table 3 shows the makespans on the test instances.

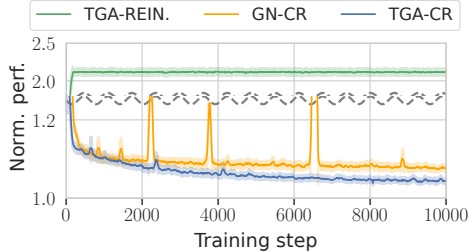

**Figure 3:** Training curves

| $N$ | $m$ | TGA-R. | GN-CR | TGA-CR |
|---|---|---|---|---|
| 100 | 5 | 6.04 | 2.67 | 2.59 |
| | 10 | 4.59 | 2.26 | 2.13 |
| | 15 | 3.56 | 2.36 | 2.07 |
| 200 | 10 | 11.27 | 2.61 | 2.45 |
| | 15 | 6.18 | 2.41 | 2.24 |
| | 20 | 4.55 | 2.52 | 2.17 |

**Table 3:** mTSP ablation results

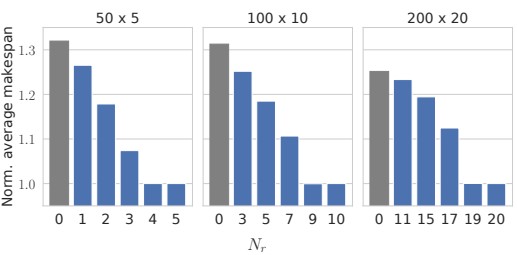

**Figure 4:** Limited observation scenarios

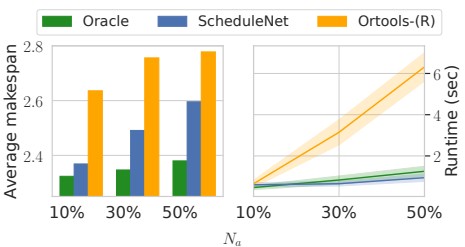

**Figure 5:** Online routing scenarios

- `TGA-REINFORCE` vs. `TGA-CR`: By comparing the learning curves of these two ablation models, we can see the effect of the proposed CR as a learning algorithm. The REINFORCE is not able to learn meaningful policy.

- `GN-CR` vs. `TGA-CR`: By comparing the learning curves of these two models, we can see the effect of TGA as a representation module. TGA induces more stabilized and faster performance improvement. In addition, as shown in Table 3, `TGA-CR` (ScheduleNet) consistently shows better performance than `GN-CR` in the test datasets. The performance difference between the two models becomes severe as $N$ and $m$ increase. This again highlights the role of TGA in solving mTSP.

### 6.3 mTSP variants experiments

In this section, we further investigate the performance of ScheduleNet in more practical scenarios of mTSP: (1) limited observation, and (2) online routing.

**Limited observation scenarios.** In the real-world application of mTSP, some salesmen may not be able to gather all the information about the other salesmen due to the limited communication capabilities (e.g., delivery trucks are located in distant). In such a scenario, the agent should decide the next city to visit with local observations. To consider this realistic scenario, we limit the number of observable salesmen $N_r$ from the global state $\mathcal{G}_\tau$ and investigate its performance.

We employ ScheduleNet to solve the test instances, whose sizes are ($50\times5$), ($100\times10$), and ($200\times20$), by varying $N_r$. As shown in Figure 4, the makespan decreases as $N_r$ increases because the extended communication scope can induce enhanced coordinations among salesmen. Note that $N_r = 0$ indicates the case where each salesman makes entirely independent actions without considering the other salesmen at all. The results implies that ScheduleNet is able to learn an effective cooperative policy that minimizes the makespan, and can perform well robustly even with limited communication capabilities.

**Online routing scenarios.** ScheduleNet, as a decentralized constructive heuristic, can solve the dynamic scheduling problem in an online manner; it reschedules inherently whenever a new event occurs (i.e., an agent finishes the assigned task or the new task appears) while considering the updated agent and task distribution. We evaluate this capability by solving the online mTSP problem where new cities arise dynamically during the course of the scheduling problem.

We first randomly generate 100 ($30\times3$) test instances. Then, while simulating these scenarios with the scheduling policies, we sequentially add $30 \times N_a\%$ cities to the simulation (the exact addition

**Table 4: Taillard's 80 results.** The gaps are measured w.r.t. the optimal (or the best-known) makespan.

| | $15\times15$ | $20\times15$ | $20\times20$ | $30\times15$ | $30\times20$ | $50\times15$ | $50\times20$ | $100\times20$ | Gap |
|---|---|---|---|---|---|---|---|---|---|
| MOR | 1.205 | 1.236 | 1.217 | 1.249 | **1.173** | 1.177 | **1.092** | 1.092 | 1.197 |
| FIFO | 1.239 | 1.314 | 1.275 | 1.319 | 1.310 | 1.206 | 1.239 | 1.136 | 1.255 |
| SPT | 1.258 | 1.329 | 1.278 | 1.349 | 1.344 | 1.241 | 1.256 | 1.144 | 1.275 |
| Zhang et al. (2020) | 1.259 | 1.300 | 1.316 | 1.319 | 1.336 | 1.224 | 1.264 | 1.136 | 1.269 |
| Park et al. (2021) | 1.171 | 1.202 | 1.228 | 1.189 | 1.254 | 1.159 | 1.174 | 1.070 | 1.181 |
| ScheduleNet | **1.153** | **1.194** | **1.172** | **1.180** | 1.187 | **1.138** | 1.135 | **1.066** | **1.154** |

moment and locations are unknown to policies before the addition) and evaluate how the policies effectively replan the schedules according to such online scenarios. We compare the makespan of ScheduleNet with (1) oracle planning that plans once while knowing all future demands (its appearing locations and times), and (2) re-planing heuristics that replans the future actions whenever $s_\tau$ is updated. Please refer to Appendix C.4 for more details regarding the simulation and baselines.

Figure 5 shows the average makespans computed using 100 random online scenarios. As shown in the left plot, ScheduleNet always produces a shorter makespan and shorter runtime than the replanning heuristic for all $N_a$ values, proving its effective adaptability and robustness to dynamically changing cities. Note that as $N_a$ increases, the makespan and the runtime increase for all methods due to the increased uncertainty and the number of replanning.

## 6.4 JSP EXPERIMENTS

We employ ScheduleNet to solve JSP, another important class of mSP, to evaluate its generalization capacity in solving various types of mSPs. The goal of the JSP is to schedule machines (i.e., agents) in a manufacturing facility to complete the jobs that consist of a series of operations (tasks) while minimizing the total completion time (makespan). Solving JSP is considered to be challenging since it imposes additional constraints that schedulers must obey: precedence constraints (i.e., an operation of a job cannot be processed until its precedence operation is done) and agent-sharing constraints (i.e., each agent has a unique set of feasible tasks).

**Formulation**. We formulate JSP as an MDP where $s_\tau$ is the partial solution of JSP, $a_\tau$ is an agent-task assignment (i.e., assigning an idle machine to one of the feasible operations), and reward is the minus of makespan. Please refer to Appendix D.1 for the detailed formulation of MDP.

**Training**. We train ScheduleNet using the random JSP instances ($N \times m$) that have $N$ jobs and $m$ machines. We sample $N$ and $m$ from $U(7, 14)$ and $U(2, 5)$ respectively to generate the training JSP instances, and randomly shuffle the order of the machines in a job to generate machine sharing constraints. Please refer to Appendix D.3 for more information.

**Results on public benchmarks**. We evaluate the makespan to verify ScheduleNet's generalization capacity to unseen JSP distributions on the Taillard's 80 dataset (Taillard, 1993). We compare ScheduleNet against the deep RL baselines (Park et al., 2021; Zhang et al., 2020), as well as three JSP heuristics; Most Operation Remaining (MOR), First-in-first-out (FIFO), and Shortest Processing Time (SPT). Both baseline RL methods were specifically designed to solve JSP by utilizing the well-known disjunctive JSP graph representation (Roy & Sussmann, 1964) and well-engineered dense reward functions. Nevertheless, we can observe that ScheduleNet outperforms all baselines in all of the cases while utilizing only a sparse episodic reward (see Table 4). Please refer to Appendix D.5 for the extended benchmark results.

## 7 CONCLUSION

In this work, we propose ScheduleNet, an RL-based scheduler that can solve various types of multi-agent scheduling problems (mSPs) in a decentralized manner. Through the extensive experiments, we empirically verified that ScheduleNet is an effective *general scheduler* that can solve various mSPs, a *cooperative scheduler* that induces multi-agent coordination to achieve a common objective, and a *scalable scheduler* that can solve large scale static scheduling problems. Furthermore, we have validated that ScheduleNet can be used to solve online scheduling problems.

## 8 ETHICS STATEMENT AND REPRODUCIBILITY DISCUSSION

**Ethics statement**   The proposed method aims to learn a decentralized policy to solve practical large-scale multi-agent scheduling problems. With the recent advanced robotics and communication technologies, the proposed method can be considered and employed as a solver for real-life logistic problems. However, our method does not consider the subjective values (social protections for the underprivileged or expedited dispatching for life-threatening demands) in the process of decision-making or learning. Thus, our method may produce scheduling results that are not well aligned with human decision-makers in such cases.

**reproducibility**   As machine learning researchers, we consider the reproducibility of numerical results as one of the top priorities. Thus, we put a significant amount of effort into pursuing the reproducibility of our experimental results. As such efforts, we set and tracked the random seed used for our experiments and confirmed the experiments were reproducible. Furthermore, we also prepare to open-source our mTSP MDP environments so that the follow-up research can be assessed in a unified way.

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

# ScheduleNet
# *Supplementary Material*

## Table of Contents

## A  DETAILS OF TYPE-AWARE GRAPH ATTENTION

In this section, we thoroughly describe the computation procedures of type-aware graph attention (TGA). Similar to the main body, We overload notations for the simplicity of notation such that the input node and edge feature as $h_i$ and $h_{ij}$, and the embedded node and edge feature $h'_i$ and $h'_{ij}$, respectively.

The proposed TGA performs graph embedding with the following three phases: (1) type-aware edge update, (2) type-aware message aggregation, and (3) type-aware node update.

**Type-aware edge update**  The edge update scheme is designed to reflect the complex type relationship among the entities while updating edge features. First, the *context* embedding $c_{ij}$ of edge $e_{ij}$ is computed using the source and destination node type $k_i, k_j$ such that:

$$c_{ij} = [\text{Emb}(k_i), \text{Emb}(k_j)] \tag{A.1}$$

where $\text{Emb}(\cdot)$ is a trainable lookup table function. Next, the type-aware edge encoding $u_{ij}$ is computed using a multilayer perceptron (MLP) as follows:

$$u_{ij} = \text{MLP}_{etype}(h_i, h_j, h_{ij}, c_{ij}) \tag{A.2}$$

where $\text{MLP}_{etype}(\cdot)$ is the type-aware edge encoding MLP. $u_{ij}$ can be seen as a dynamic edge feature which varies depending on the source and destination node type. Then, the updated edge embedding $h'_{ij}$ and its attention logit $z_{ij}$ are obtained as:

$$h'_{ij} = \text{MLP}_{edge}(u_{ij}) \tag{A.3}$$

$$z_{ij} = \text{MLP}_{attn}(u_{ij}) \tag{A.4}$$

where $\text{MLP}_{edge}$ and $\text{MLP}_{attn}$ is the edge updater and logit function, respectively. The edge updater and logit function produce updated edge embedding and logits from the type-aware edge.

The computation steps of equations A.1, A.2 and A.3 are defined as $\text{TGA}_{\mathbb{E}}$. Similarly, the computation steps of equations A.1, A.2 and A.4 are defined as $\text{TGA}_{\mathbb{A}}$.

**Type-aware message aggregation**  We first define the type-$k$ neighborhood of $v_i$ as $\mathcal{N}_k(i) = \{v_l | k_l = k, \forall v_l \in \mathcal{N}(i)\}$, where $\mathcal{N}(i)$ is the in-neigborhood set of $v_i$. The proposed type-aware message aggregation procedure computes attention score $\alpha_{ij}$ for the $e_{ij}$, which starts from $v_j$ and heads to $v_i$, such that:

$$\alpha_{ij} = \frac{\exp(z_{ij})}{\sum_{l \in \mathcal{N}_{k_j}(i)} \exp(z_{il})} \tag{A.5}$$

Intuitively speaking, The proposed attention scheme normalizes the attention logits of incoming edges over the types. Therefore, the attention scores sum up to 1 over each type-$k$ neighborhood. Next, the type-$k$ neighborhood message $m_i^k$ for node $v_i$ is computed as:

$$m_i^k = \sum_{j \in \mathcal{N}_k(i)} \alpha_{ij} h'_{ij} \tag{A.6}$$

In this aggregation step, the incoming messages of node $i$ are aggregated per type. All incoming type neighborhood messages are concatenated to produce (inter-type) aggregated message $m_i$ for $v_i$, such that:

$$m_i = \sum_{k \in \mathbb{K}} m_i^k \tag{A.7}$$

**Type-aware node update**  Similar to the edge update phase, the context embedding $c_i$ is computed first for each node $v_i$:

$$c_i = \text{Emb}(k_i) \tag{A.8}$$

where $\text{Emb}(\cdot)$ is a trainable lookup table function. Then, the updated hidden node embedding $h'_i$ is computed as below:

$$h'_i = \text{MLP}_{node}(h_i, m_i, c_i) \tag{A.9}$$

where $\text{MLP}_{node}(\cdot)$ is the type-aware node updater. The computation steps of equations A.8, and A.9 are defined as $\text{TGA}_{\mathbb{V}}$.

## B    EXTENDED DISCUSSION FOR REWARD NORMALIZATION SCHEME

In this section, we further discuss the effect of the proposed reward normalization scheme and its variants to the performance of ScheduleNet. The proposed reward normalization (i.e. normalized makespan) $m(\pi, \pi_b)$ is given as follows:

$$m(\pi, \pi_b) = \frac{M(\pi_\theta) - M(\pi_b)}{M(\pi_b)} \tag{A.10}$$

where $\pi_b$ is the baseline policy.

**Effect of the denominator**    $m(\pi, \pi_b)$ measures the relative scheduling supremacy of $\pi$ to the $\pi_b$. Similar reward normalization scheme, but without $M(\pi_b)$ division, is employed to solve single-agent scheduling problems Kool et al. (2018). We empirically found that the division leads in much stable learning when the scale of makespan change (e.g. the area of map change from the unit square to different geometries or the size of training instances are varying).

**Effect of the baseline selection**    A proper selection of $\pi_b$ is essential to assure stable and asymptotically better learning of ScheduleNet. Intuitively speaking, choosing too strong baseline (i.e. policy having smaller makespan such as LKH3 and OR-tools) can makes the entire learning process unstable since the normalized reward tends to have larger values. On the other hand, employing too weak baseline can leads in virtually no learning since the $m(\pi, \pi_b)$ becomes nearly zero.

We select $\pi_b$ as Greedy($\pi$) and this baseline selection has several advantages from selecting a fixed/pre-existing scheduling policy: (1) Entire learning process becomes independent from existing scheduling methods. Thus ScheduleNet is applicable even when the cheap-and-performing $\pi_b$ for some target mSP does not exist. (2) Greedy($\pi$) serves as an appropriate $\pi_b$ (either not too strong or weak) during policy learning. We experimentally confirmed that the baseline section Greedy($\pi$) results in a better scheduling policy as similar to the several literature Kool et al. (2018); Silver et al. (2016).

# C    DETAILS OF MTSP EXPERIMENTS

In this section, we provide the details of mTSP exepriments.

## C.1    MDP FORMULATION

The formulated mTSP MDP is event-based. Here we disccuss the further details about the event-based transitions of mTSP MDP. Whenever all agents are assigned to cities, the environment transits in time, until any of the workers arrives to the city (i.e. completes the task). Arrival of the worker to the city triggers an event, meanwhile the other assigned salesman are still on the way to their correspondingly assigned cities. We assume that each worker transits towards the assigned city with unit speed in the 2D Euclidean space, i.e. the distance travelled by each worker equals the time past between two consecutive MDP events.

It is noteworthy that multiple events can happen at the same time, typically when time stamp $t = 0$. If the MDP has multiple available workers at the same time, we repeatedly choose an arbitrary idle agent and assign it to the one of an idle task until no agent is idle, while updating event index $\tau$. This random selections do not alter the resulting solutions since we do not differentiate each agent (i.e. agents are homogeneous agents).

## C.2    AGENT-TASK GRAPH FORMULATION

In this section, we present the list of all possible node types in $\mathcal{G}_\tau$:(1) assigned-agent, (2) unassigned-agent (i.e. idle), (3) assigned-task, (4) unassigned-task, (5) inactive-task (i.e. visited city) and (6) depot. Here, we do not include inactive-agent (i.e. salesmen already return to the depot) to the graph. For edge types, we don't define them explicitly rather we consider the combination of source and destination types as the edge type. All nodes in the $\mathcal{G}_\tau$ are fully connected.

## C.3    TRAINING DETAILS

**Network parameters.**    ScheduleNet is composed of three components. The first components generates the initial node embedding $\{h_i^{(0)}\}$ and edge $\{h_{ij}^{(0)}\}$ by using linear projections. The second component (TGA), which utilizes MLP(64) as $\text{MLP}_{\text{etype}}$, $\text{MLP}_{\text{edge}}$, $\text{MLP}_{\text{attn}}$, and $\text{MLP}_{\text{node}}$, repeatedly updates the node and edge embedding 3 times to generate $\{h_i^{(3)}\}$ and edge $\{h_{ij}^{(3)}\}$. The last component $\text{MLP}_{\text{actor}}$, which is parametrized as MLP(64, 32), generates action logits from the set of node and edge embedding. All input and output dimensions of MLPs are 64 and hidden actions are LeakyReLU.

**Training pseudocode.** We presents a pseudocode for training ScheduleNet.

---

**Algorithm 2:** ScheduleNet Training

---

**input** : Training policy $\pi_\theta$, baseline policy $\pi_\phi$
**output:** Optimized policy $\pi_\theta$

1 Initialize the baseline policy with parameters $\phi \leftarrow \theta$
2 **for** *update step* **do**
3     Initialize sample buffer $\mathcal{D} \leftarrow \emptyset$
4     **for** *number of episodes E* **do**
5        Generate a random mTSP instance $I$
6        $\pi_b \leftarrow \text{Greedy}(\pi_\phi)$
7        Construct mTSP MDP from the instance $I$
8        Collect samples $\mathcal{S} = \{\mathcal{G}_\tau, a_\tau, \pi_{\theta_{\text{old}}}(a_\tau|\mathcal{G}_\tau), G_\tau(\pi_\theta, \pi_b)\}_{\tau=0}^T$ with $\pi_\theta$ and $\pi_b$ from the MDP.
9        $\mathcal{D} \leftarrow \mathcal{D} \cup \mathcal{S}$.
10     **end**
11     **for** *inner updates K* **do**
12        Calculate the loss $\mathcal{L}(\theta)$ with $\mathcal{D}$
13        $\theta \leftarrow \theta + \alpha \nabla_\theta \mathcal{L}(\theta)$
14     **end**
15     $\phi \leftarrow \beta\phi + (1 - \beta)\theta$
16 **end**
17 **return** $\pi_\theta$

---

We set the MDP discounting factor $\gamma$ to 0.99, learning rate $\alpha$ to 0.0001, $E$ to 128, $K$ to 4, $\beta$ to 0.01, and the clipping parameter $\epsilon$ of Clip-REINFORCE to 0.2. We set training seed as 1234.

### C.4 ONLINE ROUTING SCENARIOS

**Scenario generation** In this paragraph, we thoroughly explain the online routing scenario generation scheme. To generate the scenarios, we first fix the number of cities added during the online routing. Once the number of cities is given, we then decide the locations of the cities by taking midpoints of cities. We assign the location of the city that will be first added as the midpoint between the first and second cities of the mTSP map. Similarly, the location of the second city is set as the midpoint between the second and third cities. Using the same logic, we decide all positions of the newly added cities. Lastly, we decide the timing of city addition by using (near) optimal solutions (e.g., LKH3) $M^*(\pi)$. We first compute divide $M^*(\pi)$ by $1.0 < s$ and then evenly allocate the timing of additions from 0.0 to $M^*(\pi)/s$. This ensures all "to be added" cities are added during the online routing simulation no matter what scheduling policies are evaluated. We set $s = 2$.

**Baseline implementations** In this paragraph, we explain the implementations of baseline algorithms. To implement the oracle planning, we reformulate the online routing problem as a vehicle routing problem with time constraints (VRP-TW). In the VRP-TW, we set the time window constraints for the originally existing cities as $(0, M)$, where $M$ is a large number. For the newly added cities, the lower and upper bounds of the time window are set as their addition times and $M$. The re-planing heuristics are implemented with OR-tools (Perron & Furnon).

## D DETAILS OF JSP EXPERIMENTS

Job-shop scheduling problem (JSP) is a mSP that can be applied in various industries including the operation of semi-conductor chip fabrication facility and railroad system. The objective of JSP is to find the sequence of machine (agent) allocations to finish the jobs (a sequence of operations; tasks) as soon as possible. JSP can be seen as an extension of mTSP with two additional constraints: (1) precedence constraint that models a scenario where an operation of a job becomes processable only after all of its preceding operations are done; (2) agent-sharing (disjunctive) constraint that confines the machine to process only one operation at a time. Due to these additional constraints, JSP is considered to be a more difficult problem when it is solved via mathematical optimization

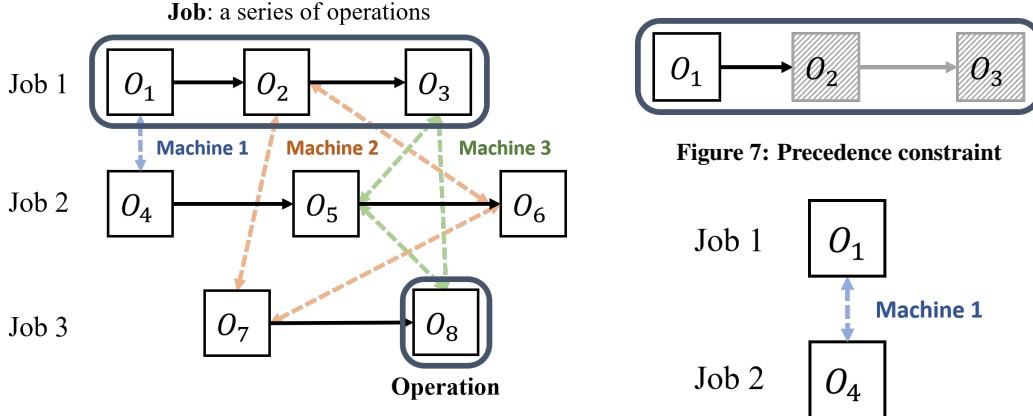

**Figure 6: Disjunctive graph representation of JSP**

**Figure 7: Precedence constraint**

**Figure 8: Agent-sharing constraint**

techniques. A common representation of JSP is the disjunctive graph representation. As shown in Figures 6, 7 and 8, JSP contains the set of jobs, machines, precedence constraints, and disjunctive constraints as its entities. In the following sections, we provide the details of the proposed MDP formulation of JSP, training details of ScheduleNet, and experiment results.

### D.1 MDP FORMULATION

The MDP formulation of JSP is similar to that of mTSP. The specific definitions of the state and action for JSP are as follows:

**State.** We define $s_\tau = (\{s_\tau^i\}_{i=1}^{N+m}, s_\tau^{\text{env}})$ which is composed of two types of states: entity state $s_\tau^i$ and environment state $s_\tau^{\text{env}}$.

- $s_\tau^i = (p_\tau^i, \mathbf{1}_\tau^{\text{processable}}, \mathbf{1}_\tau^{\text{assigned}}, \mathbf{1}_\tau^{\text{accessible}}, \mathbf{1}_\tau^{\text{waiting}})$ is the state of the $i$-th entity. $p_\tau^i$ is the processing time of the $i$-th entity at the $\tau$-th event. $\mathbf{1}_\tau^{\text{processable}}$ indicates whether the $i$-th task is processable by the target agent or not. Similar to mTSP, $\mathbf{1}_\tau^{\text{assigned}}$ indicates whether an agent/task is assigned.
- $s_\tau^{\text{env}}$ contains the current time of the environment, the sequence of tasks completed by each agent (machine), and the precedence constraints of tasks within each job.

**Action.** We define the action space at the $\tau$-th event as a set of oeprations that is both processable and currently available. Additionally, we define the *waiting* action as a reservation of the target agent (i.e. the unique idle machine) until the next event. Having *waiting* as an action allows the adaptive scheduler (e.g. ScheduleNet) to achieve the optimal scheduling solution (and also makespan) from the JSP MDP, where the optimal solution contains waiting (idle) time intervals.

### D.2 AGENT-TASK GRAPH FORMULATION

ScheduleNet constructs the *agent-task graph* $\mathcal{G}_\tau$ that reflects the complex relationships among the entities in $s_\tau$. ScheduleNet constructs a directed graph $\mathcal{G}_\tau = (\mathbb{V}, \mathbb{E})$ out of $s_\tau$, where $\mathbb{V}$ is the set of nodes and $\mathbb{E}$ is the set of edges. The nodes and edges and their associated features are defined as:

- $v_i$ denotes the $i$-th node and represents either an agent or a task. $v_i$ contains the node feature $x_i = (s_\tau^i, k_i)$, where $s_\tau^i$ is the state of entity $i$, and $k_i$ is the type of $v_i$ (e.g. if the entity $i$ is a *task* and its $\mathbf{1}_\tau^{processable} = 1$, then the $k_i$ becomes a *processable-task* type.)
- $e_{ij}$ denotes the edge between the source node $v_j$ and the destination node $v_i$. The edge feature $w_{ij}$ is a binary feature which indicates whether the destination node $v_i$ is processable by the source node $v_j$.

All possible node types in $\mathcal{G}_\tau$ are: (1) assigned-agent, (2) unassigned-agent, (3) assigned-task, (4) processable-task, and (5) unprocessable-task. We do not include completed tasks in the graph. Thus, the currently active tasks are the union of the assigned tasks, processable-tasks, and unprocessable-tasks. The full list of node features are as follows:

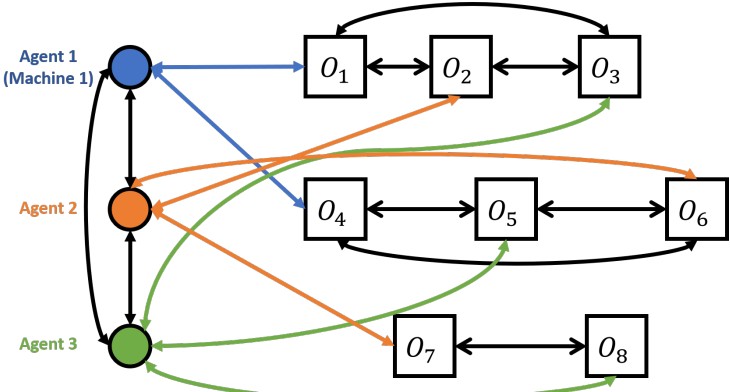

**Figure 9: JSP agent-task graph representation**

- $1_\tau^{\text{agent}}$ indicates whether the node is a agent or a task.
- $1_\tau^{\text{target-agent}}$ indicates whether the node is a target-agent (unique idle agent that needs to be assigned).
- $1_\tau^{\text{assigned}}$ indicates whether the agent/task is assigned.
- $1_\tau^{\text{waiting}}$ indicates whether the node is an agent in *waiting* state.
- $1_\tau^{\text{processable}}$ indicates whether the node is a task that is *processable* by the target-agent.
- $1_\tau^{\text{accessible}}$ indicates whether the node is processable by the target-agent and is available.
- *Task wait time* indicates the amount of time passed since the operation became *accessible*.
- *Task processing time* indicates the processing time of the operation.
- *Time-to-complete* indicates the amount of time it will take to complete the task, i.e. the time-distance to the given task.
- *Remain ops.* indicates the number of remaining operations to be completed for the job where the task belongs to.
- *Job completion ratio* is the ratio of completed operations within the job to the total amount of operations in the job.

**JSP graph connectivity**. Figure 9 visualizes the proposed agent-task graph. From Figure 9, each agent is fully connected to the set of processable tasks by that agent, and vice versa. Each task is fully connected to the other tasks (operations) that belong to the same job. Each agent is fully connected to the other agents.

## D.3 TRAINING DETAILS

We use the same training hyperparameters as in Appendix C.3.

## D.4 BASELINE IMPLEMENTATIONS

Priority dispatching rules (PDR) is one of the most common JSP solving heuristics. PDR computes the priority of the feasible operations (i.e. the set of operations whose precedent operation is done and, at the same time, the target machine is idle) by utilizing the dispatching rules. As the JSP heuristic baselines, we consider the following three dispatching rules:

- Most Operation Remaining (MOR) sets the highest priority to the operation that has the most remaining operations to finish its corresponding job.
- First-in-first-out (FIFO) sets the highest priority to the operation that joins to the feasible operation set first.
- Shortest Processing Time (SPT) sets the highest priority to the operation that has the shortest processing time.

**Table 5: Job-shop scheduling makespans on TA dataset (Part 1)**

| Instance | N × m | SPT | FIFO | MOR | Park et al. (2021) | Zhang et al. (2020) | ScheduleNet | OPT |
|----------|-------|-----|------|-----|---------------------|----------------------|-------------|-----|
| Ta01 | 15 × 15 | 1462 | 1486 | 1438 | 1389 | 1443 | 1452 | 1231 |
| Ta02 | 15 × 15 | 1446 | 1486 | 1452 | 1519 | 1544 | 1411 | 1244 |
| Ta03 | 15 × 15 | 1495 | 1461 | 1418 | 1457 | 1440 | 1396 | 1218 |
| Ta04 | 15 × 15 | 1708 | 1575 | 1457 | 1465 | 1637 | 1348 | 1175 |
| Ta05 | 15 × 15 | 1618 | 1457 | 1448 | 1352 | 1619 | 1382 | 1224 |
| Ta06 | 15 × 15 | 1522 | 1528 | 1486 | 1481 | 1601 | 1413 | 1238 |
| Ta07 | 15 × 15 | 1434 | 1497 | 1456 | 1554 | 1568 | 1380 | 1227 |
| Ta08 | 15 × 15 | 1457 | 1496 | 1482 | 1488 | 1468 | 1374 | 1217 |
| Ta09 | 15 × 15 | 1622 | 1642 | 1594 | 1556 | 1627 | 1523 | 1274 |
| Ta10 | 15 × 15 | 1697 | 1600 | 1582 | 1501 | 1527 | 1493 | 1241 |
| Ta11 | 20 × 15 | 1865 | 1701 | 1665 | 1626 | 1794 | 1612 | 1357 |
| Ta12 | 20 × 15 | 1667 | 1670 | 1739 | 1668 | 1805 | 1600 | 1367 |
| Ta13 | 20 × 15 | 1802 | 1862 | 1642 | 1715 | 1932 | 1625 | 1342 |
| Ta14 | 20 × 15 | 1635 | 1812 | 1662 | 1642 | 1664 | 1590 | 1345 |
| Ta15 | 20 × 15 | 1835 | 1788 | 1682 | 1672 | 1730 | 1676 | 1339 |
| Ta16 | 20 × 15 | 1965 | 1825 | 1638 | 1700 | 1710 | 1550 | 1360 |
| Ta17 | 20 × 15 | 2059 | 1899 | 1856 | 1678 | 1897 | 1753 | 1462 |
| Ta18 | 20 × 15 | 1808 | 1833 | 1710 | 1684 | 1794 | 1668 | 1396 |
| Ta19 | 20 × 15 | 1789 | 1716 | 1651 | 1900 | 1682 | 1622 | 1332 |
| Ta20 | 20 × 15 | 1710 | 1827 | 1622 | 1752 | 1739 | 1604 | 1348 |
| Ta21 | 20 × 20 | 2175 | 2089 | 1964 | 2199 | 2252 | 1921 | 1642 |
| Ta22 | 20 × 20 | 1965 | 2146 | 1905 | 2049 | 2102 | 1844 | 1600 |
| Ta23 | 20 × 20 | 1933 | 2010 | 1922 | 2006 | 2085 | 1879 | 1557 |
| Ta24 | 20 × 20 | 2230 | 1989 | 1943 | 2020 | 2200 | 1922 | 1644 |
| Ta25 | 20 × 20 | 1950 | 2160 | 1957 | 1981 | 2201 | 1897 | 1595 |
| Ta26 | 20 × 20 | 2188 | 2182 | 1964 | 2057 | 2176 | 1887 | 1643 |
| Ta27 | 20 × 20 | 2096 | 2091 | 2160 | 2187 | 2132 | 2009 | 1680 |
| Ta28 | 20 × 20 | 1968 | 1980 | 1952 | 2054 | 2146 | 1813 | 1603 |
| Ta29 | 20 × 20 | 2166 | 2011 | 1899 | 2210 | 1952 | 1875 | 1625 |
| Ta30 | 20 × 20 | 1999 | 1941 | 2017 | 2140 | 2035 | 1913 | 1584 |
| Ta31 | 30 × 15 | 2335 | 2277 | 2143 | 2251 | 2565 | 2055 | 1764 |
| Ta32 | 30 × 15 | 2432 | 2279 | 2188 | 2378 | 2388 | 2268 | 1784 |
| Ta33 | 30 × 15 | 2453 | 2481 | 2308 | 2316 | 2324 | 2281 | 1791 |
| Ta34 | 30 × 15 | 2434 | 2546 | 2193 | 2319 | 2332 | 2061 | 1829 |
| Ta35 | 30 × 15 | 2497 | 2478 | 2255 | 2333 | 2505 | 2218 | 2007 |
| Ta36 | 30 × 15 | 2445 | 2433 | 2307 | 2210 | 2497 | 2154 | 1819 |
| Ta37 | 30 × 15 | 2664 | 2382 | 2190 | 2201 | 2325 | 2112 | 1771 |
| Ta38 | 30 × 15 | 2155 | 2277 | 2179 | 2151 | 2302 | 1970 | 1673 |
| Ta39 | 30 × 15 | 2477 | 2255 | 2167 | 2138 | 2410 | 2146 | 1795 |
| Ta40 | 30 × 15 | 2301 | 2069 | 2028 | 2007 | 2140 | 2030 | 1669 |

## D.5 EXTENDED PUBLIC BENCHMARK JSP RESULTS

We provide the detailed JSP results for the following public datasets: TA (Taillard, 1993) (Tables 5 and 6), ORB Applegate & Cook (1991), FT Fisher (1963), YN Yamada & Nakano (1992) (Table 7), SWV Storer et al. (1992) (Table 8), and LA Lawrence (1984) (Table 9).

Table 6: Job-shop scheduling makespans on TA dataset (Part 2)

| Instance | N × m | SPT | FIFO | MOR | Park et al. (2021) | Zhang et al. (2020) | ScheduleNet | OPT |
|---|---|---|---|---|---|---|---|---|
| Ta41 | 30 × 20 | 2499 | 2543 | 2538 | 2654 | 2667 | 2572 | 2005 |
| Ta42 | 30 × 20 | 2710 | 2669 | 2440 | 2579 | 2664 | 2397 | 1937 |
| Ta43 | 30 × 20 | 2434 | 2506 | 2432 | 2737 | 2431 | 2310 | 1846 |
| Ta44 | 30 × 20 | 2906 | 2540 | 2426 | 2772 | 2714 | 2456 | 1979 |
| Ta45 | 30 × 20 | 2640 | 2565 | 2487 | 2435 | 2637 | 2445 | 2000 |
| Ta46 | 30 × 20 | 2667 | 2582 | 2490 | 2681 | 2776 | 2541 | 2004 |
| Ta47 | 30 × 20 | 2620 | 2508 | 2286 | 2428 | 2476 | 2280 | 1889 |
| Ta48 | 30 × 20 | 2620 | 2541 | 2371 | 2440 | 2490 | 2358 | 1941 |
| Ta49 | 30 × 20 | 2666 | 2550 | 2397 | 2446 | 2556 | 2301 | 1961 |
| Ta50 | 30 × 20 | 2429 | 2531 | 2469 | 2530 | 2628 | 2453 | 1923 |
| Ta51 | 50 × 15 | 3856 | 3590 | 3567 | 3145 | 3599 | 3382 | 2760 |
| Ta52 | 50 × 15 | 3266 | 3365 | 3303 | 3157 | 3341 | 3231 | 2756 |
| Ta53 | 50 × 15 | 3507 | 3169 | 3115 | 3103 | 3186 | 3083 | 2717 |
| Ta54 | 50 × 15 | 3142 | 3218 | 3265 | 3278 | 3266 | 3068 | 2839 |
| Ta55 | 50 × 15 | 3225 | 3291 | 3279 | 3142 | 3232 | 3078 | 2679 |
| Ta56 | 50 × 15 | 3530 | 3329 | 3100 | 3258 | 3378 | 3065 | 2781 |
| Ta57 | 50 × 15 | 3725 | 3654 | 3335 | 3230 | 3471 | 3266 | 2943 |
| Ta58 | 50 × 15 | 3365 | 3362 | 3420 | 3469 | 3732 | 3321 | 2885 |
| Ta59 | 50 × 15 | 3294 | 3357 | 3117 | 3108 | 3381 | 3044 | 2655 |
| Ta60 | 50 × 15 | 3500 | 3129 | 3044 | 3256 | 3352 | 3036 | 2723 |
| Ta61 | 50 × 20 | 3606 | 3690 | 3376 | 3425 | 3654 | 3202 | 2868 |
| Ta62 | 50 × 20 | 3639 | 3657 | 3417 | 3626 | 3722 | 3339 | 2869 |
| Ta63 | 50 × 20 | 3521 | 3367 | 3276 | 3110 | 3536 | 3118 | 2755 |
| Ta64 | 50 × 20 | 3447 | 3179 | 3057 | 3329 | 3631 | 2989 | 2702 |
| Ta65 | 50 × 20 | 3332 | 3273 | 3249 | 3339 | 3359 | 3168 | 2725 |
| Ta66 | 50 × 20 | 3677 | 3610 | 3335 | 3340 | 3555 | 3199 | 2845 |
| Ta67 | 50 × 20 | 3487 | 3612 | 3392 | 3371 | 3567 | 3236 | 2825 |
| Ta68 | 50 × 20 | 3336 | 3471 | 3251 | 3265 | 3680 | 3072 | 2784 |
| Ta69 | 50 × 20 | 3862 | 3607 | 3526 | 3798 | 3592 | 3535 | 3071 |
| Ta70 | 50 × 20 | 3801 | 3784 | 3590 | 3919 | 3643 | 3436 | 2995 |
| Ta71 | 100 × 20 | 6232 | 6270 | 5938 | 5962 | 6452 | 5879 | 5464 |
| Ta72 | 100 × 20 | 5973 | 5671 | 5639 | 5522 | 5695 | 5456 | 5181 |
| Ta73 | 100 × 20 | 6482 | 6357 | 6128 | 6335 | 6462 | 6052 | 5568 |
| Ta74 | 100 × 20 | 6062 | 6003 | 5642 | 5827 | 5885 | 5513 | 5339 |
| Ta75 | 100 × 20 | 6217 | 6420 | 6212 | 6042 | 6355 | 5992 | 5392 |
| Ta76 | 100 × 20 | 6370 | 6183 | 5936 | 5707 | 6135 | 5773 | 5342 |
| Ta77 | 100 × 20 | 6045 | 5952 | 5829 | 5737 | 6056 | 5637 | 5436 |
| Ta78 | 100 × 20 | 6143 | 6328 | 5886 | 5979 | 6101 | 5833 | 5394 |
| Ta79 | 100 × 20 | 6018 | 6003 | 5652 | 5799 | 5943 | 5556 | 5358 |
| Ta80 | 100 × 20 | 5848 | 5763 | 5707 | 5718 | 5892 | 5545 | 5183 |

Table 7: Job-shop scheduling makespans on ABZ, FT, ORB, and YN datasets.

| Instance | N × m | SPT | FIFO | MOR | Park et al. (2021) | ScheduleNet | OPT |
|---|---|---|---|---|---|---|---|
| abz5 | 10 × 10 | 1352 | 1467 | 1336 | 1353 | 1336 | 1234 |
| abz6 | 10 × 10 | 1097 | 1045 | 1031 | 1043 | 981 | 943 |
| abz7 | 20 × 15 | 849 | 803 | 775 | 887 | 791 | 656 |
| abz8 | 20 × 15 | 929 | 877 | 810 | 843 | 787 | 665 |
| abz9 | 20 × 15 | 887 | 946 | 899 | 848 | 832 | 678 |
| ft06 | 6 × 6 | 88 | 65 | 59 | 71 | 59 | 55 |
| ft10 | 10 × 10 | 1074 | 1184 | 1163 | 1142 | 1111 | 930 |
| ft20 | 20 × 5 | 1267 | 1645 | 1601 | 1338 | 1498 | 1165 |
| orb01 | 10 × 10 | 1478 | 1368 | 1307 | 1336 | 1276 | 1059 |
| orb02 | 10 × 10 | 1175 | 1007 | 1047 | 1067 | 958 | 888 |
| orb03 | 10 × 10 | 1179 | 1405 | 1445 | 1202 | 1335 | 1005 |
| orb04 | 10 × 10 | 1236 | 1325 | 1287 | 1281 | 1178 | 1005 |
| orb05 | 10 × 10 | 1152 | 1155 | 1050 | 1082 | 1042 | 887 |
| orb06 | 10 × 10 | 1190 | 1330 | 1345 | 1178 | 1222 | 1010 |
| orb07 | 10 × 10 | 504 | 475 | 500 | 477 | 456 | 397 |
| orb08 | 10 × 10 | 1107 | 1225 | 1278 | 1156 | 1178 | 899 |
| orb09 | 10 × 10 | 1262 | 1189 | 1165 | 1143 | 1145 | 934 |
| orb10 | 10 × 10 | 1113 | 1303 | 1256 | 1087 | 1080 | 944 |
| yn1 | 20 × 20 | 1196 | 1113 | 1045 | 1118 | 1027 | 884 |
| yn2 | 20 × 20 | 1256 | 1148 | 1074 | 1097 | 1037 | 904 |
| yn3 | 20 × 20 | 1042 | 1135 | 1100 | 1083 | 1046 | 892 |
| yn4 | 20 × 20 | 1273 | 1194 | 1267 | 1258 | 1216 | 968 |

Table 8: **Job-shop scheduling makespans on SWV datasets.**

| Instance | N × m | SPT | FIFO | MOR | Park et al. (2021) | ScheduleNet | OPT |
|---|---|---|---|---|---|---|---|
| swv01 | 20 × 10 | 1737 | 2154 | 1971 | 1761 | 1913 | 1407 |
| swv02 | 20 × 10 | 1706 | 2157 | 2158 | 1846 | 1998 | 1475 |
| swv03 | 20 × 10 | 1806 | 2019 | 1870 | 1892 | 1830 | 1398 |
| swv04 | 20 × 10 | 1874 | 2015 | 2026 | 1908 | 1971 | 1464 |
| swv05 | 20 × 10 | 1922 | 2003 | 2049 | 1796 | 1922 | 1424 |
| swv06 | 20 × 15 | 2140 | 2519 | 2287 | 2068 | 2216 | 1671 |
| swv07 | 20 × 15 | 2146 | 2268 | 2101 | 2194 | 2037 | 1594 |
| swv08 | 20 × 15 | 2231 | 2554 | 2480 | 2191 | 2255 | 1752 |
| swv09 | 20 × 15 | 2247 | 2498 | 2553 | 2278 | 2196 | 1655 |
| swv10 | 20 × 15 | 2337 | 2352 | 2431 | 2141 | 2279 | 1743 |
| swv11 | 50 × 10 | 3714 | 4427 | 4642 | 3989 | 4390 | 2983 |
| swv12 | 50 × 10 | 3759 | 4749 | 4821 | 4136 | 4532 | 2977 |
| swv13 | 50 × 10 | 3657 | 4829 | 4755 | 4008 | 4602 | 3104 |
| swv14 | 50 × 10 | 3506 | 4621 | 4740 | 3758 | 4387 | 2968 |
| swv15 | 50 × 10 | 3501 | 4620 | 4905 | 3860 | 4402 | 2885 |
| swv16 | 50 × 10 | 3453 | 2951 | 2924 | 2924 | 2924 | 2924 |
| swv17 | 50 × 10 | 3082 | 2962 | 2848 | 2840 | 2794 | 2794 |
| swv18 | 50 × 10 | 3191 | 2974 | 2852 | 2852 | 2852 | 2852 |
| swv19 | 50 × 10 | 3161 | 3095 | 3060 | 2961 | 2992 | 2843 |
| swv20 | 50 × 10 | 3125 | 2853 | 2851 | 2823 | 2823 | 2823 |

**Table 9: Job-shop scheduling makespans on LA datasets.**

| Instance | N × m | SPT | FIFO | MOR | Park et al. (2021) | ScheduleNet | OPT |
|----------|-------|-----|------|-----|--------------------|-------------|-----|
| la01 | 10 × 5 | 751 | 772 | 763 | 805 | 680 | 666 |
| la02 | 10 × 5 | 821 | 830 | 812 | 687 | 768 | 655 |
| la03 | 10 × 5 | 672 | 755 | 726 | 862 | 734 | 597 |
| la04 | 10 × 5 | 711 | 695 | 706 | 650 | 698 | 590 |
| la05 | 10 × 5 | 610 | 610 | 593 | 593 | 593 | 593 |
| la06 | 15 × 5 | 1200 | 926 | 926 | 926 | 926 | 926 |
| la07 | 15 × 5 | 1034 | 1088 | 1001 | 931 | 1008 | 890 |
| la08 | 15 × 5 | 942 | 980 | 925 | 863 | 863 | 863 |
| la09 | 15 × 5 | 1045 | 1018 | 951 | 951 | 951 | 951 |
| la10 | 15 × 5 | 1049 | 1006 | 958 | 966 | 958 | 958 |
| la11 | 20 × 5 | 1473 | 1272 | 1222 | 1276 | 1254 | 1222 |
| la12 | 20 × 5 | 1203 | 1039 | 1039 | 1039 | 1039 | 1039 |
| la13 | 20 × 5 | 1275 | 1199 | 1150 | 1150 | 1150 | 1150 |
| la14 | 20 × 5 | 1427 | 1292 | 1292 | 1292 | 1292 | 1292 |
| la15 | 20 × 5 | 1339 | 1587 | 1436 | 1282 | 1395 | 1207 |
| la16 | 10 × 10 | 1156 | 1180 | 1108 | 1134 | 1047 | 945 |
| la17 | 10 × 10 | 924 | 943 | 844 | 953 | 888 | 784 |
| la18 | 10 × 10 | 981 | 1049 | 942 | 1049 | 947 | 848 |
| la19 | 10 × 10 | 940 | 983 | 1088 | 880 | 963 | 842 |
| la20 | 10 × 10 | 1000 | 1272 | 1130 | 1042 | 989 | 902 |
| la21 | 15 × 10 | 1324 | 1265 | 1251 | 1309 | 1261 | 1046 |
| la22 | 15 × 10 | 1180 | 1312 | 1198 | 1158 | 1027 | 927 |
| la23 | 15 × 10 | 1162 | 1354 | 1268 | 1085 | 1145 | 1032 |
| la24 | 15 × 10 | 1203 | 1141 | 1149 | 1129 | 1088 | 935 |
| la25 | 15 × 10 | 1449 | 1283 | 1209 | 1308 | 1117 | 977 |
| la26 | 20 × 10 | 1498 | 1372 | 1411 | 1553 | 1458 | 1218 |
| la27 | 20 × 10 | 1784 | 1644 | 1566 | 1624 | 1516 | 1235 |
| la28 | 20 × 10 | 1610 | 1474 | 1477 | 1438 | 1357 | 1216 |
| la29 | 20 × 10 | 1556 | 1540 | 1437 | 1582 | 1320 | 1152 |
| la30 | 20 × 10 | 1792 | 1648 | 1565 | 1649 | 1490 | 1355 |
| la31 | 30 × 10 | 1951 | 1918 | 1836 | 1817 | 1906 | 1784 |
| la32 | 30 × 10 | 2165 | 2110 | 1984 | 1977 | 1850 | 1850 |
| la33 | 30 × 10 | 1901 | 1873 | 1811 | 1795 | 1731 | 1719 |
| la34 | 30 × 10 | 2070 | 1925 | 1853 | 1895 | 1784 | 1721 |
| la35 | 30 × 10 | 2118 | 2142 | 2064 | 2041 | 1969 | 1888 |
| la36 | 15 × 15 | 1799 | 1516 | 1492 | 1489 | 1449 | 1268 |
| la37 | 15 × 15 | 1655 | 1873 | 1606 | 1623 | 1653 | 1397 |
| la38 | 15 × 15 | 1404 | 1475 | 1455 | 1421 | 1444 | 1196 |
| la39 | 15 × 15 | 1534 | 1532 | 1540 | 1555 | 1430 | 1233 |
| la40 | 15 × 15 | 1476 | 1531 | 1358 | 1570 | 1357 | 1222 |

