# OpenReview forum: "ScheduleNet: Learn to solve multi-agent scheduling problems with reinforcement learning"
_ICLR.cc/2022/Conference — ICLR 2022 Submitted_

### Official Review · Reviewer_6uYb · 2021-11-02

**Correctness:** 3
**Technical Novelty And Significance:** 2
**Empirical Novelty And Significance:** 2
**Recommendation:** 6
**Confidence:** 3

**Main Review:**

Strengths: The proposed ScheduleNet can make a tradeoff between computation time and solution efficiency (i.e., makespan).
Weaknesses:
1)	From the experimental results, I find the proposed ScheduleNet less attractive. Specially, in the large-scale instance (200, 20), the LKH3 can also return the optimal solution. Although the running time of LKH3 is nearly 1,000 seconds, the ScheduleNet also takes hundreds of seconds. Considering the solution gap between LKH3 and ScheduleNet is 1.04, it is worth of waiting LKH3, especially in the offline settings, where the agents and jobs are known.
2)	The JSP task seems easier than the mTSP task, since the machine in JSP is homogeneous while the salesman in mTSP is heterogeneous with different positions. Thus, my question is that why unify the training frame for these two different scenarios? I am convinced that the ScheduleNet is suitable for mTSP, but more smart mechanism is necessary for JSP. This can also explain that the ScheduleNet performs worse in JSP (Table 4) than that in mTSP. A potential re-organization is to first propose a more simple mechanism for the JSP with homogenous agents, and then to propose a complex mechanism (e.g., ScheduleNet) for the mTSP.
3)	In section 4.1, agent-task graph, the edge e_{ij} between nodes should be defined more clearly. Especially when there should be an edge between two nodes?
4)	This paper claims to propose a coordinated multi-agent approach, ScheduleNet seems like a centralized training and execution framework. Although the authors state “Note that ScheduleNet allows an agent to process its local state information,….appear dynamically.”, it is not clear how these agents coordinate their behavior, for example, when two agents are assigned to the same job, which agent really executes it?
5)	Typos: Section 3: “An event is defined as the the case”->“An event is defined as the case”.


**Summary Of The Paper:**

This paper proposes a multi-agent reinforcement learning approach to solve scheduling problems (including mTSP and job-shop scheduling (JSP)), with the objective of makespan minimization. In the mTSP, the agent represents salesman, and in JSP, the agent represents machine. The core idea is to employ the type-aware graph attention mechanism to learn the assignment probability of agents to tasks. Experiments on mTSP and JSP simulations validate the strength of the proposed approach ScheduleNet.

**Summary Of The Review:**

This paper proposes a multiagent reinforcement learning approach to solve scheduling problems (including mTSP and job-shop scheduling (JSP)). However, the techniques are not clearly described and the experimental results are not fascinating.

---

> ### Author Response · Authors · 2021-11-16
> **Reply to the Reviewer 6uYb**
>
> Thank you for reviewing our paper and the effort for the review. Here we provide the explanation to your comments.
>
> **About the efficacy of ScheduleNet**
>
> We humbly admit that our algorithm's performance is not superb compared to algorithms designed to solve only specific problems. However, we still believe that the proposed algorithm has its strong merits in that it is a general scheduling algorithm that can be used for solving the various types of scheduling problems. In addition, the performance gap is also due to the different ways of formulation for solving target scheduling problems. That is, LKH3 is a centralized approach, while ScheduleNet is a constructive decentralized approach. We seek to have a decentralized problem-solving strategy to achieve the transferability of the algorithm to unseen problems and scalability for solving large-scale problems with a reasonable computational time.
>
> In addition, we believe that the applicability and usefulness of an algorithm should be evaluated in the context where the algorithm is employed to solve specific tasks. For example, if one needs to compute the near-optimum scheduling solution while using enormous computing resources and computational time, LKH3 would be preferable. However, on the contrary, if one needs to find a reasonably good scheduling solution in a relatively short amount of time for dealing with time-critical tasks, such as patient transportation, vaccine delivery, and food delivery, an algorithm with a faster execution time will be preferable.
>
> Furthermore, when solving dynamic or stochastic vehicle routing or machine scheduling problems, one must repeatedly re-compute the entire scheduling solutions. In such a case, LKH3 will spend an enormous amount of computing time. On the contrary, ScheduleNet will just use the same computational time as the static scheduling problems. In addition, in such a dynamic and stochastic environment, finding the optimum solution computed while assuming that one knows all the future events is not that important and meaningful. Instead, rapidly calculating a reasonably good re-plan is a lot more critical and practical. We empirically show the trained ScheduleNet can cope well with those online routing scenarios from 6.3.
>
> **About the selections of mTSP and JSP as the testing problems.**
>
> Because mTSP and JSP are seemingly different types of problems, it may seem unreasonable to solve them using a unified sequential decision-making framework, ScheduleNet. However, it is meaningful to identify the structural characteristics common to these seemingly different scheduling problems and propose a unified and generalized approach to solve these problems while reflecting the identified common structural properties.
>
> The common aspects that we found from these two problems and our approaches to reflect them are as follows:
>
> - We need to have a state representation that is flexible enough to represent the relationships among any arbitrary number of agents and tasks.
> - We need to introduce the coordination among multiple agents to complete the geographically distributed tasks as quickly as possible using a sequential and decentralized decision-making strategy
> - We need to learn such a decentralized cooperative policy using only a delayed and sparse reward signal, makespan, which is revealed only at the end of the episode.
>
> We believe the performance differences between mTSP and JSP are originated from the difficulty of the problems themselves. Both mTSP and JSP are often cast as mixed Integer programming (MIP). However, the MIP formulation of TSP has a relatively smaller number of constraints than the one of JSP. For example, JSP has the job-precedence constraints and the machine sharing constraints. Such complicated constraints hinder the construction of optimum solutions since even a tiny mistake can induce infeasible solutions or severely-performance-degraded solutions. In general, the difficulty of solving MIP increase exponentially with the number of constraints.
>
> **About the edge connectivity in mTSP**
>
> We construct the agent-task graph as the complete graph following the TSP (or vehicle routing problem) conventions as mentioned in section 4.1. We update the main text to more clearly explain it.

---

> > ### Author Response · Authors · 2021-11-16
> > **Reply to the Reviewer 6uYb (Continued)**
> >
> > **About the coordination among the agents**
> >
> > This study proposes a learning-based decentralized and cooperative sequential decision-making algorithm for solving multi-agent scheduling problems. ScheduleNet is a decentralized policy in that the trained single decentralized policy maps local observation of each idle salesman one of feasible individual action. Decentralization of scheduling policy is essential to ensure the learned policy can be employed to schedule any size of mTSP problems in a scalable manner. In addition, cooperation is required to ensure that all agents coordinate to minimize the total completion time (global team reward). This cooperative behavior is induced by setting the shared team reward among agents when training a decentralized policy.
> >
> > Typical multi-agent reinforcement learning (MARL) for the cooperative game tries to derive decentralized policies for agents that can improve the global (team) objective of the agents (solving Decentralized MDP (Dec-MDP)). In Dec-MDP, the MDP transitions are made when all agents' actions are concurrently exerted to the environment. Thus, the concurrency is the source of non-stationarity as the agents can't observe the other's actions when making decisions. One employs the centralized training and decentralized execution (CTDE) learning concept to effectively derive the decentralized cooperative policy while overcoming this nonstationary issue.
> >
> > However, our formulation for a multi-agent scheduling problem (mSP) is different from a typical dec-MDP. We formulate mSP as an event-based MDP to model the asynchronous action-based transition; an event occurs whenever an agent reaches the target customer and becomes idle, and the action taken by this idle agent induces the state transition. In this formulation, an idle agent (vehicle) can observe other agents' actions (ongoing traveling to the target cities) when making its decision. Such event-based asynchronous transition eliminates conflicting action between agents. In addition, this observation can be thought of as implicit communication among agents and facilitate cooperation among agents.
> >
> > **About the typo**
> >
> > We will update the typo. Thank you for your thorough review.
> >
> > **About the experimental results**
> >
> > In terms of performance, ScheduleNet is the SOTA solver among RL-based solvers. Although there are not many (multi-agent) RL-based algorithms for solving multi-agent scheduling problems, we have verified that ScheduleNet performs better than [1,2] in mTSP and performs better than [3,4] in JSP. We agree that the performance of ScheduleNet is not superior to search-based solvers that exhaustively search over the entire solution space. However, if we consider the computational time, ScheduleNet can be regarded as SOTA, especially for large-scale problems.
> >
> > [1] Yujiao Hu, Yuan Yao, and Wee Sun Lee. A reinforcement learning approach for optimizing multiple traveling salesman problems over graphs. Knowledge-Based Systems, 204:106244, 2020.
> >
> > [2] Cao, Yuhong, Zhanhong Sun, and Guillaume Sartoretti. "DAN: Decentralized Attention-based Neural Network to Solve the MinMax Multiple Traveling Salesman Problem." *arXiv preprint arXiv:2109.04205* (2021).
> >
> > [3] Cong Zhang, Wen Song, Zhiguang Cao, Jie Zhang, Puay Siew Tan, and Chi Xu. Learning to dispatch for job shop scheduling via deep reinforcement learning. 34th Conference on Neural Information Processing Systems (NeurIPS 2020).
> >
> > [4] Park, Junyoung, et al. "Learning to schedule job-shop problems: representation and policy learning using graph neural network and reinforcement learning." International Journal of Production Research 59.11 (2021): 3360-3377.

---

> ### Author Response · Authors · 2021-12-02
> **To Reviewer 6uYb**
>
> We did our best to answer your questions. Unfortunately, however, we have so far not received any comment from the reviewer on our responses and corrections. We hope that our response provides sufficient information for you to re-evaluate this manuscript. We've summarized the update here in case you missed it.
>
> - **About the efficacy of ScheduleNet:** We believe that the applicability and usefulness of an algorithm should be evaluated in the context where the algorithm is employed to solve specific tasks. As we provided in the comments, for time-critical usage or dynamic scenario, ScheduleNet can be a better choice than LKH3.0.
> - **About the selections of mTSP and JSP as the testing problems:** We select mTSP and JSP as the targeting multiple agent scheduling problems as they share similar technical difficulties to derive the learnable schedulers. We provide the details in the comments.
> - **About the coordination among the agents:** We design ScheduleNet architecture as it can serve as a decentralized policy in that the trained single decentralized policy maps local observation of each idle salesman one of feasible individual action. In addition, we train ScheduleNet with global team rewards so that cooperative behaviors are induced. Additionally, to handle MARL's non-stationary issues, we design our MDP can handle the common multi-agent problems. In our MDP, whenever each agent makes a decision, it can observe the other agents and their actions. This property leaves out the non-stationary issues.
> - **About the experimental results:** We showed the performance of ScheduleNet is state-of-the-art among the RL-based construction heuristics for both mTSP and JSP.
>
> It would be greatly appreciated if you would re-evaluate our paper based on our responses and revised manuscript or provide additional questions/comments.
>
> Sincerely,
>
> The authors.

---

### Official Review · Reviewer_7p6U · 2021-11-02

**Correctness:** 3
**Technical Novelty And Significance:** 3
**Empirical Novelty And Significance:** 3
**Recommendation:** 6
**Confidence:** 3

**Main Review:**

==== A post rebuttal edit: I am satisfied with the author's extended answer, therefore I raise my score.

The paper is an interesting application of RL coupled with some tricks to the well-known optimization problems. Below are the detailed remarks.

​1. Description of MDP:

 * $p_\tau^i$ is defined as 'position' in Section 3.1. This is not clear. Section C.1. suggests that positions are 2D points.​
 * The generative process of MDP should be explained: how are $p_\tau^i$ generated, what is the distribution of distances between 'positions'? This could help to give context to the numbers in Tables 1-4.​
 * In the context of the above, the definition of makespan should be made more explicit.

2) Training:
* PPO uses an advantage normalization trick (among others). It bears some resemblance to the normalization trick in the paper (which is on the reward side, though). Is there some deeper similarity here?​
* Using advantage makes the method invariant to scalar shift in the rewards. This seems not to be the case for normalized rewards used in the paper. Is this issue a limitation for the method to use it for different optimization problems?  ​
* It is not clear how the formula (9) is computed, in particular using data gathered in Algorithm 2. This should be made explicit. Similarly, the meaning of the expected value in formula (9) should be described.​
* It would be interesting to know why the choice of "slowly varying" policy as baseline helps. In particular, it seems that this choice has to be related to the *inner updates* loop from Algorithm 2 (lines 9-12) and the choice of $K$. It also somewhat resembles the use of target networks in contemporary RL algorithms.
  ​
3) Experiments:

* The description of training in Section 6.1 is slightly confusing. Is the ScheduleNet trained on $30x3$ setup for both Table 1 and Table 2? What is the choice of the *update steps* (lines 2 in Algorithm 2)?​
* It seems from Appendix C.1. that only one seed is used for training (1234). If so, then this could be worrying, as RL training is known to have high variance.​
* Values from tables 1-4 are not clear:
    - what are the entries (makespan? normalized? discounted?)?
    - can we interpret these numbers (see the questions in point 1)?
    - what are the confidence intervals for these results?
    - how does this result compares to the "size of the graph", e.g. its
      diameter?
    - lack of the above make is if difficult to assess the difference between
      ScheduleNet and the baselines.​
* "Gap" (last column in Tables 1, 2, 4) is not defined. Make sure that the entries in this column reflect the values computed according to the definition and the values in the table.​
* The Authors mention real-life applications; however do not specify their scale.
  In particular, are the problems in the experiments section close to real
  applications?
  ​
* What are the computational limits of the method? How much larger problems can
  be tackled larger computational resources?

**Summary Of The Paper:**

The paper considers solving two well-known combinatorial optimization problems (multiple traveling salesmen problem and job-shop scheduling problem) using neural networks trained with the help of Reinforcement Learning. The approach embeds the problem in an MDP framework with rewards corresponding to a normalized makespan metric and trains the policy using a PPO-like objective.

**Summary Of The Review:**

The paper is an interesting application of RL coupled with some tricks to the
well-known optimization problems.
In its current form, it is below the acceptance threshold, but I am willing
to increase my score if the authors address the issues raised above.

---

> ### Author Response · Authors · 2021-11-16
> **Reply to the Reviewer 7p6U**
>
> We really appreciate your sincere review of our manuscript. The followings are the clarifications and answers to your comments and questions. We hope that our responses help to clarify our works. We will also update the manuscript based on your comments to more clearly explain our works.
>
> **About the description of MDP**
> - For mTSP, $p_\tau^i$ denotes the Cartesian coordinate of $i$-th entity at the $\tau$-th event. That is, it represents the location of either a vehicle, a customer, or the depot depending on 2D space.
> - The x, y components of $p_0^i$'s (i.e. initial coordinates of the mTSP entities) are sampled from Uniform distribution whose lower and upper bounds are 0 and 1, respectively, as explained 6.1 Training.
> - The makespan is defined as the total completion time for all agents to finish all the tasks and return to the depot. Thus, it is determined by the last agent that returns to the depot the last. As we assume the unit traveling speed of salesmen, the makespan is equivalent to the minimal length sub tour among all the sub tours completed by agents (the makespan is the minimum tour length).
>
> After reading the review, current Section 3.1 may not be straightforward for all readers to follow. We will revise the writing and try to explain the MDP formulation as clearly as possible. In addition, we will leave a note for guiding the readers to the appendix.
>
> **About training**
> - **The similarity between proposed training tricks and PPO:** From the perspective of decreasing the variance of returns, our reward normalization trick is similar to the PPO2 implementation [1] using the advantage normalization trick.
>
> - **Justification of the proposed normalization scheme:** We have used the proposed normalization scheme (Eq. 7) in an attempt to train a single policy that can solve any mTSP instance with any number of agents (M) and any number of customers (N). This is a type of multi-task learning in that we attempt to derive a single policy for conducting multiple tasks with different goals (i.e., different city layouts with different agents). In training a single policy using the various sized mTSP instances, it is essential to objectively evaluate the current policy's performance. If the reward signal's scale changes a lot due to the size of the sampled training mTSP instances, the policy parameters cannot be trained stably. For this reason, we normalize the makespan reward signal to reduce the variance of the reward signal caused by the randomness of the mTSP training instance. Moreover, this normalization changes the definition of reward. Hence, it doesn't violate the unbiasedness of the policy gradient estimator.
>
> [1] https://stable-baselines.readthedocs.io/en/master/_modules/stable_baselines/ppo2/ppo2.html#PPO2
>
> **Clarification about the Equation (9) and algorithm chart 2.**
> Sorry for your confusion due to other misspecification of one variable.
>
> $\rho_\tau=\pi_\theta(a_\tau|g_\tau)/\pi_{\color{red}\theta_\text{old}}(a_\tau|g_\tau)$
>
> where $\pi_{\color{red}\theta_\text{old}}(a_\tau|g_\tau)$ is the probability of selecting action $a_\tau$ at $g_\tau$ when policy $\pi_\theta$ is used. Note that we use $g_\tau$ to denote the state (graph) of $\tau$-th event since the markdown do not render "\mathcal{G}".
>
> The detailed procedure for calculating Equation 9 is explained as follow:
> - Collecting the samples: Assume we have current policy $\pi_\theta$ and baseline policy $\pi_b$ at hand. We run the two policies on the same mSP instance and compute the makespans for the two policies, i.e., $M(\pi_\theta)$, $M(\pi_b)$, by solving the mSP with  $\pi_\theta$ and $\pi_b$, respectively. While solving mSP, we gather the state $g_\tau$, action $a_\tau$, and $\pi_{\theta_\text{old}}(a_\tau|g_\tau)$ trajectory that has been generated by following $\pi_\theta$.
>
> - Calculate the normalized returns: We compute the normalized reward as stated in Equation (7). Later we calculate the normalized return $G_\tau(\pi_\theta,\pi_b)$ using the equation (8) for all state-action pairs $(g_\tau, a_\tau)$.
>
> - Calculate $\mathcal{L}(\theta)$: We can then estimate the performance of policy for a sampled $(\mathcal{G}_\tau, a_\tau)$ pair as follows: $\mathcal{L}_\theta(\mathcal{G}_\tau, a_\tau)=\min(\text{clip}(\rho_\tau,1-\epsilon,1+\epsilon)G_\tau, \rho G_\tau)$ where $
> \text{clip}(x,a,b)= \begin{cases}
> a, \text{if}  x < a \\
> x, \text{if}  a \leq x \leq b \\
> b, \text{if}  x > b \\
> \end{cases}$. By computing the empirical mean of the $\mathcal{L}_\theta(\mathcal{G}_\tau, a_\tau)$ for the sampled pairs, as stated in equation (9), we can estimate $\mathcal{L}(\theta)$ more precisely.
>
> We will revise the manuscript to more clearly explain the procedure to compute the loss function and its gradient in the main text. In addition, we will also update Algorithm 2 in the Appendix to clearly explain the learning procedure.

---

> > ### Author Response · Authors · 2021-11-16
> > **Reply to the Reviewer 7p6U (Continued)**
> >
> > **Further explanation about the "slowly varying" baseline**
> >
> > The primary rationale for employing the “slowly varying” baseline is to use the consistent reward signal to update the policy $\pi_\theta$. Since we train a single policy using random mTSP instances with different numbers of agents and customers, using unnormalized makespan as a reward signal will prevent consistent and stable learning (because the scale of makespan for different sized problems varies a lot). Thus, reward normalization is essential to standardize the reward signal.
> >
> > Then, the remaining question is how to normalize the reward signal. We decide to use the previous policy as the baseline policy
> > $\pi_b$ to normalize the reward, i.e., $\frac{M(\pi_\theta) -M(\pi_b)}{M(\pi_b)}$, where $\pi_b$ is the previous version of  $\pi_\theta$. This scheme induces the updating policy $\pi_\theta$ to perform better than its previous version $\pi_b$. Also, this selection was inspired by common RL tricks; soft target functions.
> >
> > The **inner update** routine is directly adopted from the PPO's update routine. As the clipped objective function is weighed by $\frac{\pi_\theta(a_\tau|g_\tau)}{ \pi_{\color{red}\theta_\text{old}}(a_\tau|g_\tau)}$, the updating policy $\pi_\theta(a_\tau|g_\tau)$ is regularized to be similar to the old policy $\pi_{\theta_\text{old}} (a_\tau|g_\tau)$. It allows we can perform multiple rounds of parameter updates, up-to-certain degree, without recollecting the samples in on-policy RL without decreasing policy performance
> >
> > **About experiments:**
> >
> > **Clarification of Section 6.1**
> >
> > In this study, we train a single ScheduleNet policy using random mTSP instances sampled from the uniform distribution. We denote ($N \times m$) as the mTSP with $N$ cities (tasks) and $m$ salesmen (agents). To generate a random mTSP instance, we sample $N$ and $m$ from $U(15, 30)$ and $U(3, 4)$, respectively.
> > Similarly, the Euclidean coordinates of N cities are sampled uniformly at random from the unit
> > square. We use this ScheduleNet to solve the random mTSP instances with different N and m combinations (Table 1) and the realistic mTSP instances in mTSPLib (Table 2).
> >
> > We choose $K$=4 which is the default hyperparameter of PPO2 as described in C.3.
> >
> > **About the fixed random seed**
> >
> > We used one ScheduleNet model, which is trained on the explained hyperparameter setup and random seed 1234, to produce the results of Table 1,2 and Figure 4,5. We also tested the ScheduleNet model with different random seeds in Section 6.2 to produce the results of **TGA-CR** of Figure 3.
> >
> > **Entries of the tables**
> > The numerical metrics used in all the tables are summarized below:
> > - The numbers in Table 1 except "gap" indicate the average makespan computed by applying the target algorithm to solve 100 test mTSP instances. For each combination of  $n$ and $m$, all the methods solve the same 100 target mTSP instances sampled from the uniform distribution. The average makespan $\bar M_d(\pi)=\frac{1}{100}\sum_{i=1}^{100} M_d(e_i;\pi)$ where $M(e_i;\pi)$ is the makespan of $\pi$ for $i$-th instance of the dataset $d$. The numbers in the rightmost column indicate the gap representing the relative performance of the target algorithm compared to the LKH. The gap is defined as $\frac{1}{D}\sum_{d=1}^{D}\frac{\bar M_d(\pi)}{\bar M_d(\pi_{\text{LKH}})}$.
> > - The entries in Tables 2 are makespan values computed by applying the target algorithm to a specific test mTSP instance.  The rightmost columns provide the Gap computed similar to Table 1.
> > - The number in Table 3 also shows the average makespan values computed from the specific mTSP instances of size $(N\times m)$.
> > - Table 4 reports the average makespan gaps measured w.r.t. the optimal (or the best-known) makespan, following the report style of one of our Deep-RL baselines [5]. the average makespan is computed as $\frac{1}{I}\sum_{i=1}^{I}\frac{ M(e_i,\pi)}{\bar M^*(e_i)}$ where $e_i$ is the $i$-th JSP instance in the dataset and $M^*(e_i)$ is the optimal makespan.
> >
> > **Reporting Confidence Interval**
> >
> > Here, we provide the sample standard deviations of the makespan values computed when applying the proposed model and other baseline models to 100 randomly generated mTSP instances (all models solve the same set of 100 mTSP test instances). The confidence interval can then be computed using these sample standard deviations.

---

> > > ### Author Response · Authors · 2021-11-16
> > > **Reply to the Reviewer 7p6U (Continued)**
> > >
> > > |Size|LKH|Ortools|ScheduleNet (g.)|ScheduleNet (s.64)|
> > > |:---:|:---:|:---:|:---:|:---:|
> > > |30 x 3|2.17 $\pm$ 0.184|2.25 $\pm$ 0.190|2.32 $\pm$ 0.184|2.22 $\pm$ 0.187|
> > > |30 x 5|1.94 $\pm$ 0.278|1.95 $\pm$ 0.272|2.02 $\pm$ 0.257|1.96 $\pm$ 0.275|
> > > |50 x 5|2.00 $\pm$ 0.228|2.04 $\pm$ 0.217|2.17 $\pm$ 0.214|2.07 $\pm$ 0.224|
> > > |50 x 7|1.95 $\pm$ 0.247|1.96 $\pm$ 0.244|2.07 $\pm$ 0.235|1.99 $\pm$ 0.239|
> > > |50 x 10|1.91 $\pm$ 0.312|1.91 $\pm$ 0.322|1.98 $\pm$ 0.298|1.92 $\pm$ 0.302|
> > > |100 x 5|2.20 $\pm$ 0.196|2.35 $\pm$ 0.173|2.59 $\pm$ 0.218|2.43 $\pm$ 0.201|
> > > |100 x 10|1.97 $\pm$ 0.289|2.03 $\pm$ 0.337|2.13 $\pm$ 0.249|2.03 $\pm$ 0.262|
> > > |100 x 15|1.98 $\pm$ 0.275|2.03 $\pm$ 0.308|2.07 $\pm$ 0.250|1.99 $\pm$ 0.266|
> > > |200 x 10|2.04 $\pm$ 0.266|2.23 $\pm$ 0.497|2.45 $\pm$ 0.245|2.25 $\pm$ 0.227|
> > > |200 x 15|2.00 $\pm$ 0.294|2.34 $\pm$ 0.521|2.24 $\pm$ 0.254|2.08 $\pm$ 0.264|
> > > |200 x 20|1.97 $\pm$ 0.279|2.37 $\pm$ 0.530|2.17 $\pm$ 0.252|2.05 $\pm$ 0.268|
> > >
> > > All models have a similar level of standard deviation, and it is hard to find clear relationships between the level of standard deviation and the size of target problems.
> > >
> > > **About the size of graph**
> > >
> > > We generate a graph following the convention of the current research field (RL to solve vehicle routing problems). We create graphs with different numbers of nodes in 1 X 1 unit square. In addition, we use the fully connected graph, meaning that every node is connected with each other. The makespan generally increases with the number of customers since vehicles should travel more to visit all the customers. In contrast, the makespan decreases typically with the number of agents since more agents mean smaller duty per single agent. However, these trends are not always consistent since the slight change in the relative locations can sometimes affect a lot to the final makespan (due to the combinatorial nature of the routing problems).
> > >
> > > **Are the problems in the experiments section close to real applications?**
> > >
> > > We thought that for a scheduling algorithm to be used for solving real-world applications, the scheduler should be able to re-plan the scheduling actions corresponding to dynamically changing scenarios.
> > >
> > > ScheduleNet, as a decentralized constructive heuristic, can solve the dynamic scheduling problem in an online manner. That is, it can inherently reschedule whenever a new event occurs, i.e., an agent finishes the assigned task or the new task appears while considering the updated agent and task distribution.  We have validated that ScheduleNet is more effective in replacing compared to the re-planing heuristics.
> > >
> > > Even though we verified the planning capability using the maps whose size is less than 200, we can easily employ the trained algorithm to larger problems with more customers. Because ScheduleNet is a decentralized constructive policy, there is no limitation in the size of the target problem, although the execution time increases, and the performance might decrease as the size of the test problem increase.
> > >
> > > The following table shows that how the performance of the ScheduleNet changes as the number of salesmen and cities dramatically increases. We used the trained ScheduleNet model trained on N and m from $U(15, 30)$ and $U(3, 4)$ respectively, without any further training.
> > >
> > >
> > > |Size|LKH|Ortools|ScheduleNet (g.)|
> > > |:---:|:---:|:---:|:---:|
> > > |300 x 30|2.05 $\pm$ 0.294|2.94 $\pm$ 0.709|2.25 $\pm$ 0.257
> > > |500 x 50|2.06 $\pm$ 0.275|7.55 $\pm$ 0.594|2.31 $\pm$ 0.312
> > > |750 x 50|2.07 $\pm$ 0.261|10.90 $\pm$ 0.684|2.50 $\pm$ 0.293
> > >
> > > As we can notice, ScheduleNet maintains relatively good performance even for such large-sized problems.
> > >
> > > **About the computational limits**
> > >
> > > The training scheme of ScheduleNet is more efficient than other deep-RL-based scheduling algorithms, AM[6] and DAN[7]. While other algorithms require tracking gradient over the entire MDP episode, our proposed method can be trained using every state-action data. This difference of training allows ScheduleNet to be trained more effectively with large problem instances.
> > >
> > > Our bottleneck for computational resources is the memory bandwidth for training GNN. The memory footprint of GNN increases roughly with $\mathcal{O}(LE^2)$, where $L$ is the number of GNN layers and E is the number of edges when we assume to use a complete graph. In our experimental setup, the peak GPU memory usage was ~12GB with the provided hyperparameters in the appendix. However, we can overcome the limits by employing distributed training within the GNN as in large graph problems. Thus, we expect the proposed method to scale much larger training cases while enjoying the distributed GNN training on multiple GPU settings.
> > >
> > > [6] Kool, Wouter, Herke Van Hoof, and Max Welling. "Attention, learn to solve routing problems!.
> > > [7] Cao, Yuhong, Zhanhong Sun, and Guillaume Sartoretti. "DAN: Decentralized Attention-based Neural Network to Solve the MinMax Multiple Traveling Salesman Problem.

---

> ### Author Response · Authors · 2021-11-23
> **We are wondering if our responses addressed your questions well.**
>
> We have answered most of your questions and we are wondering if our responses addressed your questions well. If your questions or concerns were resolved, we are wondering if you can consider raising the score, as you mentioned.
>
> Thank you again for your precious comments.

---

> ### Author Response · Authors · 2021-12-02
> **To Reviewer 7p6U**
>
> Thanks for updating the score. We are so glad that the provided reply explain your comments well.
>
> Sincerely,
>
> The authors.

---

### Official Review · Reviewer_XP6V · 2021-11-04

**Correctness:** 3
**Technical Novelty And Significance:** 3
**Empirical Novelty And Significance:** 4
**Recommendation:** 6
**Confidence:** 4

**Details Of Ethics Concerns:**

The major ethical concerns are explained well in the paper.

**Main Review:**

The paper proposes a generic framework for solving multi-agent combinatorial scheduling problems. The type-aware embedding for agent-task relationship graph construction is novel and has a distributed framework which can be developed using message passing framework among neighbours. The reward normalization and usage of clip parameter in training REINFORCE algorithm are shown to be effective. I appreciate the extensive amount of empirical results on both mTSP and JSP domains, which demonstrate that the ScheduleNet not only performs efficiently during training process as compared to state-of-the-art methods, but also generalizes well to other real-world datasets. Ablation study also demonstrates the efficacy of embedding methods and two practical modifications made for REINFORCE algorithm. Having said that I have a few comments: (a) The runtime comparison between RL and OR methods is not completely fair as the RL method only consider inference time (training time is not included if I am not wrong). (b) The multi-agent aspect of the problem is somewhat not discussed in properly. I would appreciate adding a distributed RL training algorithm using the messages generated for each node in each time-step. One key issue in multi-agent RL training is non-stationarity; I am not sure how this problem is tackled. (c) Please explain how the different hard constraints for these combinatorial problems are satisfied during RL training? Is it generalizable enough to tackle additional constraints, e.g., time window constraints? (d) I am interested to understand the problem setting for the results of Table 2. The RL algorithm is trained using synthetic dataset (results of Table 1) and then executed in these public datasets? Have you done any incremental training on these public datasets? If not, do you have any insights on why it generalizes so well to these public datasets; does the parametric distribution matches between the synthetic and public dataset? Please clarify.

**Summary Of The Paper:**

The paper proposes a generic approach called ScheduleNet to solve combinatorial multi-agent scheduling problems using neural network and reinforcement learning. An agent-task relationship graph is first constructed through edge and node embedding (using message passing framework among neighbours). Message values for each node is computed using type-aware edge embedding value and then a node embedding is performed using the message values. Using these embedding values, ScheduleNet computes assignment probability between idle agent and active tasks. ScheduleNet is trained using reinforcement learning and two methods are proposed to improve training performance: (a) reward normalization and (b) adding clip parameter in REINFORCE algorithm training. Extensive experimental results are demonstrated on mTSP and job-shop scheduling problems. Through benchmarking against state-of-the-art OR heuristics and RL methods, and using ablation studies, it is shown that the ScheduleNet performs efficiently and generalizes well to unknown or online scenarios.

**Summary Of The Review:**

The paper solves a combinatorial multi-agent scheduling problem using graph embedding and RL methods. Both type-aware message embedding and practical suggestions to improve REINFORCE algorithm are interesting and performs well. I also appreciate extensive empirical results on two general problem classes against both state-of-the-art OR and RL based techniques. Having said that, I am not confident on how the multi-agent aspect of the problem is tackled. No theoretical result is provided to demonstrate that the ScheduleNet can handle all the hard domain constraints of a problem and can tackle the multi-agent interactions in a principled way.

---

> ### Author Response · Authors · 2021-11-16
> **Reply to the Reviewer XP6V**
>
> **(a) Runtime comparisons**
>
> As you pointed out the training time is not counted in "runtime" of Figure 2. However, the training is required once. We used one trained ScheduleNet to solve all mTSP problems reported in Sections 6.1, 6.2, and 6.3.
>
> In the perspective of practitioners employing ScheduleNet in solving target scheduling problems, the execution time for generating a good solution is an essential factor. That is why we only compare the runtimes in this paper. It is also noteworthy that reporting the only runtimes is a convention in the research field applying RL to solve combinatorial optimization problems.
>
> Lastly, we also like to mention that training of ScheduleNet can be done in a reasonable time. In our desktop computer equipped with AMD Threadripper 2990WX CPU and RTX Titan GPU, the entire training was done ~24 Hours.
>
> **(b) Discussion about the multi-agent aspect of the target problem.**
>
> This study proposes a learning-based decentralized and cooperative sequential decision-making algorithm for solving multi-agent scheduling problems. Decentralization of scheduling policy is essential to ensure the learned policy can be employed to schedule any size of mTSP problems in a scalable manner. The trained single decentralized policy maps local observation of each idle salesman one of feasible individual action. In addition, cooperation is required to ensure that all agents coordinate to minimize the total completion time. This coordination is induced by using the global team reward shared by all agents.
>
> Typical multi-agent reinforcement learning (MARL) for the cooperative game tries to derive decentralized policies for agents that can improve the global (team) objective of the agents (solving Decentralized MDP — Dec-MDP). In Dec-MDP, the MDP transitions are made when all agents' actions are concurrently exerted to the environment. Thus, the concurrency is the source of non-stationarity as the agents can't observe the other's actions when making decisions. One employs the centralized training and decentralized execution (CTDE) learning concept to effectively derive the decentralized cooperative policy while overcoming this nonstationary issue.
>
> However, our formulation for a multi-agent scheduling problem (mSP) is different from a typical dec-MDP. We formulate mSP as an event-based MDP to model the asynchronous action-based transition; an event occurs whenever an agent reaches the target customer and becomes idle, and the action taken by this idle agent induces the state transition. In this formulation, an idle agent (vehicle) can observe other agents' actions (ongoing travels to the target cities) when making its decision. Therefore, the nonstationary issue does not occur.
>
> The differences between these two problem classes, "dec-MDP", and "event-based dec-MDP", are discussed in [1].
>
> [1] Becker et al., "Decentralized Markov Decision Processes with Event-Driven Interactions," In ****Proceedings of the Third International Joint Conference on Autonomous Agents and Multiagent Systems, 2004. AAMAS 2004.
>
> **(c) Generalizability of the proposed MDP to the different combinatorial problems**
>
> As a learned construction heuristic, ScheduleNet builds a solution sequentially by assigning an idle agent to one of the remaining tasks while considering the relationships among the remaining tasks and agents. This construction heuristic ensures that the learned policy can assign an idle agent only a feasible action, eliminating the need for explicit modeling and enforcing expensive constraints (e.g., sub-tour eliminations of mTSP).
>
> The proposed ScheduleNet framework is general enough to tackle any additional constraints. The only thing we need to do is eliminate the infeasible actions while considering the constraint information imposed in the target problem and have the policy choose one of the remaining feasible actions at every event-based transition. For example, we can exclude the infeasible selection at the current state considering the imposed time constraints to consider time-window constraints. Of course, the constraints information is provided as part of the state to model how the constraints affect optimum scheduling actions properly.
>
> Lastly, we would like to mention that ScheduleNet has already been used to solve a job-shop scheduling problem (JSP) with more complex constraints, precedence constraints, and machine sharing constraints and has proven to solve such scheduling problems effectively.

---

> > ### Author Response · Authors · 2021-11-16
> > **Reply to the Reviewer XP6V (Continue)**
> >
> > **(d) About the test setup of mTSP public benchmarks**
> >
> > We trained only a single ScheduleNet using random mTSP instances. To generate a random mTSP instance, we sample N and m from the uniform distributions, U(15, 30) and U(3, 4), respectively. Similarly, the Euclidean coordinates of N cities are sampled uniformly at random from the unit square. ScheduleNet is trained on these random mTSP instances that are generated on the fly.
> >
> > To investigate the generalization capacity of ScheduleNet to various problem sizes, we evaluate ScheduleNet on the randomly generated mTSP datasets. Each (N × m) dataset consists of 100 random uniform mTSP instances (Table 1). In addition, to explore the generalization of ScheduleNet to problems that come from entirely different distributions (i.e., real-world data), we employ the trained ScheduleNet to solve mTSP instances in the mTSPLib [1]. These results are provided in Table 2. The following two figures show the two representative maps, eli51-2 and berlin 52-2, from mTSPLib. In those maps, ScheduleNet shows better scheduling performance than DAN (deep RL baseline). As we can notice from the maps, especially berlin52, the cities are distributed far from uniform. So, it is not true that ScheduleNet attains good performance in mTSPLib because the training and the test mTSP instances are similar (i.e., similar data distributions).
> >
> > **[ELI-BERLIN IMAGE LINK](https://drive.google.com/file/d/1h6Iog4cYrNwtrytMXBbCZ7bgwSN7c7PR/view?usp=sharing)**
> >
> > ScheduleNet shows good zero-shot transferability. The reason for such good generalizations is due to (1) general graph representation for scheduling problems (Section 4.1) and (2) policy learning using reinforcement learning with stabilizing techniques.
> >
> > - General graph representation: ScheduleNet employs a message-passing-based graph neural network while utilizing edge features to capture relative locations among agents and tasks effectively. On the other hand, most RL approaches for solving combinatorial optimization employ a transformer architecture while utilizing only the coordinates of agents and tasks. It is known that using the edge feature is effective in solving shortest path problems (e.g., Dijkstra / Bellman-Ford algorithm).
> > - Policy learning using reinforcement learning with stabilizing techniques: ScheduleNet uses the normalized makespan reward to universally evaluate the algorithm's performance while solving various problems with different sizes.
> >
> > Thus, the representation and policy learning strategies are synergized to induce good generalization.
> >
> > [1] https://profs.info.uaic.ro/~mtsplib/MinMaxMTSP/index.html

---

> ### Author Response · Authors · 2021-12-02
> **To Reviewer XP6V**
>
> We did our best to answer your questions. Unfortunately, however, we have so far not received any comment from the reviewer on our responses and corrections. We hope that our response provides sufficient information for you to re-evaluate this manuscript. We've summarized the update here in case you missed it.
> - **About the runtimes:** we report the inference times (i.e., roll out the learned policy) as the runtimes. Training takes some amount of time. However, it is noteworthy that training ScheduleNet from scratch can be done ~ 24 hours with a desktop computer.
> - **About the consideration of the multi-agent aspect of the target problem:** we design our MDP can handle the common multi-agent problems. In our MDP, whenever each agent makes a decision, it can observe the other agents and their actions. This property leaves out the non-stationary issues.
> - **About the generalizability of the proposed method:** we design MDP so that the sequence of states is the solution constructing sequence. This idea can be extended to a more complex scenario. For instance, to model CVRP with time window constraints, we can make an agent (vehicle) select the tasks (city) that meet window constraints by considering different action spaces. Our Job-shop scheduling problem can be seen as one extension of mTSP with more complex constraints.
> - **About the zero-shot transferability to the different data distribution:** we conjecture that the reason for such good generalizations is due to (1) general graph representation for scheduling problems (Section 4.1) and (2) policy learning using reinforcement learning with stabilizing techniques.
>
> It would be greatly appreciated if you would re-evaluate our paper based on our responses and revised manuscript or provide additional questions/comments.
>
> Sincerely,
>
> The authors.

---

### Official Review · Reviewer_VLpo · 2021-11-09

**Correctness:** 3
**Technical Novelty And Significance:** 2
**Empirical Novelty And Significance:** 3
**Recommendation:** 6
**Confidence:** 3

**Main Review:**

----A post rebuttal edit:
Thanks for providing the detailed clarifications, a revised manuscript, and new experimental results. I have read all these responses. I agree with the advantages of ScheduleNet over other RL-based methods; however, I’m just still not convinced about the novelty of the three modules themselves. Besides, the explanation about not using the value function is far-fetched, and there is already much research about assignments in the MARL community. In addition, when comparing with other RL-based methods, I suggest that the authors provide the number of network parameters to ensure that the advantages achieved by ScheduleNet do not depend on a larger network. However, I keep my positive impression on this work.


**Strengths**:

1. The paper is well-organized, and the writing logic is very clear. The Introduction section does not give a flat description but introduces the features of the proposed method one by one; therefore, the readers can have a quick glance at the advantages of the ScheduleNet. The Method section is also clearly structured and easy to understand.
2. The reinforcement learning framework is suitable for solving sequential decision problems, so it is a reasonable and clever attempt to apply the framework to multi-agent scheduling problems. This paper's adjustments in modeling, representation, and training techniques are all good practices to involve the RL framework.

**Weaknesses**:

1. This submission has some mistakes and typos in writing, including grammar and expression problems, some of which may affect understanding. I only list some of them here.

   (1) Verb mistakes. For example, in the last paragraph on page one, "a wrong choice at the early stage can results (result) in irreversible poor results at the end"; In the part about Job-shop scheduling problems on page three, "However, these methods utilizing (utilize) human-engineered dense reward"......

   (2) Third-person singular. For example, "learning-based improvement (improvements) have shown better performance" in the paragraph about vehicle routing problems on page two; "such randomness do (does) not alter" in the action paragraph on page three;

   (3) Writing standard mistakes. For example, nearly all the "i.e." miss the comma (i.e.,);

   (4) Compilation problems. For example, Figure 1 on page three is mentioned on page four for the first time;

   (5) Many article usage mistakes. For example, you should remove the word "the" of "following novelties and the advantages" in the Novelties paragraph on the second page; In the paragraph about Job-shop scheduling problems on page three, "solving job-shop scheduling problem" should be revised as "solving the job-shop scheduling problem" or "solving job-shop scheduling problems"; Besides, the "an" in "requires an additional training" should be removed; There are two "the" in the transition paragraph on page three;

   (6) Sentence break mistakes. For example, "A solution to mTSP is considered to be complete when all the cities have been visited (, ) and all salesmen have returned to the depot" in the first paragraph of page four, a comma should be put before "and" because the subject of the sentence changes;

   (7) ......

   While some of these errors do not affect comprehension, they may influence the cogency of this paper. Besides, these basic errors may give the impression that the essay has not been adequately prepared.

2. There are linguistic repetitions in the formulation section. The authors try to give examples to clearly explain some of the definitions but mostly just repeat the definitions—for example, the reward part in subsection 3.1. If the paper sets a subsection about examples, it is recommended to give a straightforward and real example rather than repetition.

3. Why does the essential difference between min-max and min-sum affect the solutions to scheduling problems? This point is understandable but not theoretically stated in the paper. Therefore, the reader cannot tell whether the proposed framework can also solve the min-sum problem. I believe that different scheduling problems may pursue various goals, and the problem with min-max as the goal that this paper tries to solve is only one class of scheduling problems. If the proposed method cannot solve problems with min-sum as the goal, then the proposed approach may have some limitations. I suggest that the authors analyze and explain these issues in their submission.

4. About Novelty. The approach proposed in this paper is an attempt to apply reinforcement learning to scheduling problems. The modeling technique, representation technique, and RL technique are minor improvements on traditional methods and can be summarized as implementation tricks to adapt to scheduling problems. I am therefore skeptical about this paper's novelty and contribution. In terms of the RL community, the contribution of this paper is low.

   I have not previously delved into scheduling problems and have only a passing knowledge in this area, so I cannot accurately assess this method's academic and theoretical contributions in the scheduling area. However, by reading this submission, I believe that its theoretical contribution may be small and the experimental performance is not significantly improved compared to other methods.

   However, if the authors can provide more convincing explanations during the discussion period, I will change my opinion on this problem.

**Summary Of The Paper:**

This paper tries to solve traditional scheduling problems by using the deep reinforcement learning (RL) framework. To make the RL framework applicable to such problems, the authors model the problem states as an agent-task graph and encode the nodes in the graph utilizing a type-aware graph attention technique they proposed. The obtained node embeddings are next used to compute the final assignment probabilities. The authors use implementation tricks such as reward normalization and clip-loss to accelerate training or achieve better performance in the training process. The proposed method performs well in the traveling salesmen problem and the job-shop scheduling problem.

**Summary Of The Review:**

This paper is an attempt to combine the DRL framework with scheduling problems. It deserves credit for modeling the scheduling problems as MDPs to fit the RL framework and for providing new implementation insights for solving mSPs by utilizing reinforcement learning. However,  this paper has no major innovations or new theoretical insights at the methodological level but minor changes to existing techniques to make a REINFORCE variant applicable to the scheduling problem. Therefore, this paper's contribution to the community may be limited.

---

> ### Author Response · Authors · 2021-11-16
> **Reply to the Reviewer VLpo**
>
> We really appreciate your thorough review comments. The followings are our responses to your questions and comments.
>
> **About Improving the writing**
>
> We agree that there were many grammatical errors and problems with wording. We will do our best to correct these mistakes and improve the quality of writing.
>
> **About the linguistic repetitions**
>
> Thank you very much for pointing out the mistakes we made over and over again unconsciously. We will pay attention to these repeated mistakes and try to improve the writing style, especially for Section 3.1.
>
> **About the applicability of different scheduling objectives**
>
> We seek to develop a cooperative scheduling policy that can minimize the total completion time (makespan) of the distributed tasks for two reasons: (1) minimizing the makespan (min-max) is the most direct performance metric in many time-critical applications (application-oriented goal) than min-sum problems, and (2) training decentralized cooperative policies using only the makespan, a delayed and sparse episodic reward, is notoriously difficult (academic-oriented goal).
>
> As the reviewer stated, the min-max scheduling problem is one of the various scheduling problems; however, the min-max scheduling problem is a general modeling framework to determine the sequence of multiple agents for minimizing the total completion time of the entire task. For example, it can be used to schedule various time-critical tasks, such as vaccine delivery, emergent patient management, and defense operations. The min-sum problem will be a good modeling approach if, for example, the company needs to minimize the vehicles' fuel consumption instead of minimizing the time. Because we believe that the min-max problem is general enough to cover the diversified practical problems, our method designed to solve this min-max problem is not limited.
>
> The advantage of the RL framework is that it can induce different policies to achieve any objective if the proper reward signal is used for the target objective. In that sense, ScheduleNet can be still used without any difficulty to solve the min-sum problem if we use the total sum of the traveling distance of agents as the final reward. The state representation learning module based on TGA can still be used without any change. To prove that ScheduleNet can be used to solve min-sum scheduling problems, we have trained ScheduleNet using the min-sum reward (i.e., the total traveling distance). Except for the reward signal, we use the same representation module and the hyperparameters as the min-max case during training. The trained policy is then used to all min-sum scheduling problems.
>
> *Examples of LKH minmax and minsum solution*
>
> **[MINMAX vs. MINSUM IMAGE LINK](https://drive.google.com/file/d/1tciqYnHoJEfMoqN4PzSPXzkGyBfyDStC/view?usp=sharing)**
>
> To clearly show the difference between the min-max and min-sum formulation, we show the near-optimum solutions for these two problems solved by the LKH method. The left figure shows the (near) optimum routing results for the min-sum problem, and the right figure shows the near-optimum routing results for the min-max problem. In the min-max problem, the tasks are distributed to all agents to minimize the total completion time. On the other hand, in the min-sum problem, only a fraction of agents visit the customers while other agents are idle; this solution is effective in minimizing the total traveling distance, and its solution is almost similar to the solution of solving TSP with a single agent. From this example, we can clearly understand the different solution tendencies between these two problem classes.
>
> *Min-sum results*
>
> |Size|LKH|Ortools|ScheduleNet (g.)|ScheduleNet (s.64)|
> |:---:|:---:|:---:|:---:|:---:|
> |30 x 3|4.88 $\pm$ 0.374|4.56 $\pm$ 0.301|4.82 $\pm$ 0.365|4.72 $\pm$ 0.342|
> |30 x 5|5.60 $\pm$ 0.490|4.61 $\pm$ 0.259|4.88 $\pm$ 0.303|4.75 $\pm$ 0.267|
> |50 x 5|6.54 $\pm$ 0.392|5.86 $\pm$ 0.285|6.25 $\pm$ 0.314|6.09 $\pm$ 0.294|
> |50 x 7|7.19 $\pm$ 0.605|5.82 $\pm$ 0.299|6.25 $\pm$ 0.359|6.10 $\pm$ 0.308|
> |50 x 10|8.45 $\pm$ 0.717|5.82 $\pm$ 0.262|6.23 $\pm$ 0.322|6.09 $\pm$ 0.303|
> |100 x 5|8.28 $\pm$ 0.333|8.06 $\pm$ 0.293|8.83 $\pm$ 0.400|8.58 $\pm$ 0.306|
> |100 x 10|9.73 $\pm$ 0.639|8.06 $\pm$ 0.317|8.76 $\pm$ 0.352|8.51 $\pm$ 0.291|
> |100 x 15|11.52 $\pm$ 1.112|8.09 $\pm$ 0.306|8.84 $\pm$ 0.352|8.56 $\pm$ 0.321|
> |200 x 10|11.99 $\pm$ 0.411|11.22 $\pm$ 0.274|12.76 $\pm$ 0.568|12.29 $\pm$ 0.383|
> |200 x 15|13.31 $\pm$ 0.657|11.17 $\pm$ 0.269|12.76 $\pm$ 0.548|12.30 $\pm$ 0.342|
> |200 x 20|14.91 $\pm$ 0.890|11.19 $\pm$ 0.238|12.80 $\pm$ 0.458|12.40 $\pm$ 0.350|
>
> From these results, we have empirically proven that the ScheduleNet framework can derive the mTSP scheduling policy for minimizing the min-sum objective as well. This series of experiments highlights the potential of the ScheduleNet framework for handling various scheduling objectives.

---

> > ### Author Response · Authors · 2021-11-16
> > **Reply to the Reviewer VLpo (Continued)**
> >
> > **About the novelty**
> >
> > We propose ScheduleNet not only to achieve the best performance but also to address the more challenging research question “how to design a learning-based solve to solve large-scale scheduling problems in a decentralized and cooperative manner.”
> >
> > In this study, we propose a learning-based decentralized and sequential decision-making algorithm for solving multi-agent scheduling problems; the trained policy, a construction heuristic, can be employed to solve scheduling problems with any number of agents and tasks.
> >
> > Learning a transferable solver in a construction heuristic framework is significantly challenging compared to its single-agent variants (e.g., TSP solver) because (1) we need to use the state representation that is flexible enough to represent any arbitrary number of agents and tasks (2) we need to introduce the coordination among multiple agents to complete the geographically distributed tasks as quickly as possible using a sequential and decentralized decision making strategy, and (3) we need to learn such decentralized cooperative policy using only a delayed and sparse reward signal, makespan, that is revealed only once at the end of the episode.
> >
> > To tackle such a challenging task, we formulate multiple agent scheduling problems (mSP) as an event-based MDP and derive a decentralized decision-making policy in an RL framework using only a sparse and delayed episodic reward signal. The major technical novelties are summarized again as follows:
> >
> > - **(Formulation) Decentralized cooperative decision-making strategy**:
> > Decentralization of scheduling policy is essential to ensure the learned policy can be employed to schedule any size of mSP in a scalable manner; decentralized policy maps local observation of each idle salesman one of feasible individual action while joint policy mapa the global state to the joint scheduling actions.
> > - **(Representation) State representation via agent-task graph and type-aware graph attention (TGA)**:
> > The proposed method represents a state (a partial solution of mSP) as an agent-task graph that captures specific relationships among agents and tasks which turns out to be essential in our ablation study. ScheduleNet then employs TGA to compute the node embeddings for all nodes (tasks and agents), which are used to assign an idle salesman to an unfinished task.
> > - **(Training) Training decentralized policy using a single delayed shared reward signal:** Training decentralized cooperative strategy using a single sparse and delayed reward is extremely difficult in that we need to distribute credits of a single scalar reward (makespan) over time and agents. To resolve this, we propose a stable RL training scheme that significantly stabilizes the training and improves the generalization performance.
> >
> > In terms of performance, ScheduleNet is the SOTA solver among RL-based ones. Although there are not many (multi-agent) RL-based algorithms for solving multi-agent scheduling problems, we have verified that ScheduleNet performs better than [1,2] in mTSP and performs better than [3,4] in JSP. We agree that the performance of ScheduleNet is not superior to search-based solvers that exhaustively search over the entire solution space. However, if we consider the computational time, ScheduleNet can be regarded as SOTA, especially for large-scale problems.
> >
> > [1] Yujiao Hu, Yuan Yao, and Wee Sun Lee. A reinforcement learning approach for optimizing multiple traveling salesman problems over graphs. Knowledge-Based Systems, 204:106244, 2020.
> > [2] Cao, Yuhong, Zhanhong Sun, and Guillaume Sartoretti. "DAN: Decentralized Attention-based Neural Network to Solve the MinMax Multiple Traveling Salesman Problem." *arXiv preprint arXiv:2109.04205* (2021).
> > [2] Cong Zhang, Wen Song, Zhiguang Cao, Jie Zhang, Puay Siew Tan, and Chi Xu. Learning to dispatch for job shop scheduling via deep reinforcement learning. 34th Conference on Neural Information Processing Systems (NeurIPS 2020).
> > [4] Park, Junyoung, et al. "Learning to schedule job-shop problems: representation and policy learning using graph neural network and reinforcement learning." International Journal of Production Research 59.11 (2021): 3360-3377.

---

> ### Author Response · Authors · 2021-12-02
> **To Reviewer VLpo**
>
> We did our best to answer your questions. Unfortunately, however, we have so far not received any comment from the reviewer on our responses and corrections. We hope that our response provides sufficient information for you to re-evaluate this manuscript. We've summarized the update here in case you missed it.
>
> - We revised and corrected the sentences of the manuscript. The updates are highlighted in blue in the revised manuscript.
> - We removed the linguistic repetitions in the formulation section.
> - We empirically proved that the proposed ScheduleNet solves **minsum** TSP well. The results are shared in our comments.
> - We summarized the novelties (or academic contributions) of the proposed method in our comments. In short, each part of the proposed method (i.e., problem formulation, graph representation, its processing unit — TGA —, and training scheme) is devised to solve the targeting multiple agent scheduling problem well. We backed up this claim through our numerical experiments — SOTA among the RL approaches —, ablation studies and applicability of the proposed method to Job-shop scheduling problems.
>
> It would be greatly appreciated if you would re-evaluate our paper based on our responses and revised manuscript or provide additional questions/comments.
>
> Sincerely,
>
> The authors.

---

### Author Response · Authors · 2021-11-19
**To all reviewers**

We sincerely thank all reviewers for the careful reviews and inputs. We make some updates to the manuscript so that the revised version can fix the issues found during the review period. For the updated parts, we highlight the sentences with blue colors.

---

### Decision · Program_Chairs · 2022-01-20

**Decision:**

Reject

**Comment:**

This paper proposes a deep RL framework for the traditional schedule problem. The proposed algorithm is shown to be effective and has zero-shot generalization abilities. Reviewers are mostly satisfied with the response and the overall evaluation is slightly positive. However, there are some drawbacks of the current paper preventing it from getting a higher evaluation: (1) The reviewers believe that the contribution might be small -- at least for the RL area; the experimental performance for the scheduling problem is also not significantly improved compared to other methods (e.g. the search-based ones). Hence the reviewers believe the contribution of the paper is limited. (2) There is a number of typos and language issues in its present version. The paper may need several rounds of polishment before publication. (3) There is a lack of theoretical justification for the proposed method.  In sum, the AC recommends a borderline rejection.